# Kernelized Reinforcement Learning with Order Optimal Regret Bounds

**Sattar Vakili**
MediaTek Research
Cambridge, UK
sattar.vakili@mtkresearch.com

**Julia Olkhovskaya**
TU Delft*
Delft, the Netherlands
julia.olkhovskaya@gmail.com

## Abstract

Reinforcement learning (RL) has shown empirical success in various real world settings with complex models and large state-action spaces. The existing analytical results, however, typically focus on settings with a small number of state-actions or simple models such as linearly modeled state-action value functions. To derive RL policies that efficiently handle large state-action spaces with more general value functions, some recent works have considered nonlinear function approximation using kernel ridge regression. We propose $\pi$-KRVI, an optimistic modification of least-squares value iteration, when the state-action value function is represented by a reproducing kernel Hilbert space (RKHS). We prove the first order-optimal regret guarantees under a general setting. Our results show a significant polynomial in the number of episodes improvement over the state of the art. In particular, with highly non-smooth kernels (such as Neural Tangent kernel or some Matérn kernels) the existing results lead to trivial (superlinear in the number of episodes) regret bounds. We show a sublinear regret bound that is order optimal in the case of Matérn kernels where a lower bound on regret is known.

## 1   Introduction

Reinforcement learning (RL) in real world often has to deal with large state action spaces and complex unknown models. While RL policies using complex function approximations have been empirically effective in various fields including gaming (Silver et al., 2016; Lee et al., 2018; Vinyals et al., 2019), autonomous driving (Kahn et al., 2017), microchip design (Mirhoseini et al., 2021), robot control (Kalashnikov et al., 2018), and algorithm search (Fawzi et al., 2022), little is known about theoretical performance guarantees in such settings. The analysis of RL algorithms has predominantly focused on simpler cases such as tabular or linear Markov decision processes (MDPs). In a tabular setting, a regret bound of $\tilde{\mathcal{O}}(\sqrt{H^3|\mathcal{S} \times \mathcal{A}|T})$ has been shown for optimistic state-action value learning algorithms (e.g., see, Jin et al., 2018), where $H$ is the length of episodes, $T$ is the number of episodes, and $\mathcal{S}$ and $\mathcal{A}$ are finite state and action spaces. This bound does not scale well when the size of state-action space grows large. Furthermore, when the model (the state-action value function or the transitions) admits a $d$-dimensional linear representation in some state-action features, a regret bound of $\tilde{\mathcal{O}}(\sqrt{H^3d^3T})$ is established (Jin et al., 2020), that scales with the dimension of the linear model rather than the cardinality of the state-action space.

Several recent studies have explored the utilization of complex models with large state-action spaces. A very general model entails representing the state-action value function using a reproducing kernel Hilbert space (RKHS). This approach allows using kernel ridge regression to obtain confidence intervals, which facilitate the design and analysis of RL algorithms. The most significant contribution

---

*Work was done when the author was affiliated with Vrije Universiteit Amsterdam.

37th Conference on Neural Information Processing Systems (NeurIPS 2023).

to this general RL problem is Yang et al. (2020a),[2] that provides regret guarantees for an optimistic least-squares value iteration (LSVI) algorithm, referred to as kernel optimistic least-squares value iteration (KOVI). The main assumption is that the state-action value function can be represented using the RKHS of a known kernel $k$. The regret bounds reported in Yang et al. (2020a) scale as $\tilde{\mathcal{O}}\left(H^2\sqrt{(\Gamma(T) + \log\mathcal{N}(\epsilon))\,\Gamma(T)T}\right)$, with $\epsilon = \frac{H}{T}$, where $\Gamma(T)$ and $\mathcal{N}(\epsilon)$ are two kernel related complexity terms, respectively, referred to as maximum information gain and $\epsilon$-covering number of the class of state-action value functions. The definitions are given in Section 4. Both complexity terms are determined using the spectrum of the kernel. While for smooth kernels, characterized by exponentially decaying Mercer eigenvalues, such as Squared Exponential kernel, $\Gamma(T)$ and $\log\mathcal{N}(\frac{H}{T})$ are logarithmic in $T$, for more general kernels with greater representation capacity, these terms may grow polynomially in $T$, possibly making the regret bound trivial (superlinear).

To have a better understanding of the existing result, let $\{\sigma_m > 0\}_{m=1}^\infty$ denote the Mercer eigenvalues of the kernel $k$ in a decreasing order. Also, let $\{\phi_m\}_{m=1}^\infty$ denote the corresponding eigenfeatures. Refer to Section 2.2 for details. The kernel $k$ is said to have a polynomial eigendecay when $\sigma_m$ decay at least as fast as $m^{-p}$ for some $p > 1$. The polynomial eigendecay profile satisfies for many kernels of practical and theoretical interest such as Matérn family of kernels (Borovitskiy et al., 2020) and the Neural Tangent (NT) kernel (Arora et al., 2019). For a Matérn kernel with smoothness parameter $\nu$ on a $d$-dimensional domain, $p = \frac{2\nu+d}{d}$ (e.g., see, Janz et al., 2020). For a NT kernel with $s - 1$ times differentiable activations, $p = \frac{2s-1+d}{d}$ (Vakili et al., 2021b). In Yang et al. (2020a), the regret bound is specialized for the class of kernels with polynomially decaying eigenvalues, by bounding the complexity terms based on the kernel spectrum. However, the reported regret bound is sublinear in $T$ only when the kernel eigenvalues decay very fast. In particular, let $\tilde{p} = p(1 - 2\eta)$, where for $\eta \geq 0$, $\sigma_m^\eta \phi_m$ is uniformly bounded. Then, Yang et al. (2020a), Corollary 4.4 reports a regret bound of $\tilde{\mathcal{O}}(T^{\xi^* + \kappa^* + \frac{1}{2}})$, with

$$\kappa^* = \max\{\xi^*, \frac{2d+p+1}{(d+p)(\tilde{p}-1)}, \frac{2}{\tilde{p}-3}\}, \quad \xi^* = \frac{d+1}{2(p+d)}. \tag{1}$$

The regret bound $\tilde{\mathcal{O}}(T^{\xi^* + \kappa^* + \frac{1}{2}})$ is sublinear only when $p$ and $\tilde{p}$ are sufficiently large. That, at least, requires $2\xi^* < \frac{1}{2}$, implying $p > d + 2$, when $\tilde{p}$ is also sufficiently large. For instance, for Matérn kernels, this requirement can be expressed as $\nu > \frac{d(d+1)}{2}$, when $\frac{(2\nu+d)(1-2\eta)}{d}$ is sufficiently large.

**Special case of bandits.** A similar issue existed in the simpler problem of kernelized bandits, corresponding to the special case where $H = 1, |\mathcal{S}| = 1$. Specifically, the $\tilde{\mathcal{O}}(\Gamma(T)\sqrt{T})$ regret bounds reported for optimistic sampling (Srinivas et al., 2010a, GP-UCB), as well as for Thompson sampling (Chowdhury and Gopalan, 2017, GP-TS) are also trivial (superlinear) when $\Gamma(T)$ grows faster than $\sqrt{T}$. It remains an open problem whether the suboptimal performance guarantees for these two algorithms is a fundamental shortcoming or an artifact of the proof. This observation is formalized as an open problem on the online confidence intervals for RKHS elements in Vakili et al. (2021d). For the kernelized bandits problem, Scarlett et al. (2017) proved lower bounds on regret in the case of Matérn family of kernels. In particular, they proved an $\Omega(T^{\frac{\nu+d}{2\nu+d}})$ lower bound on regret of any bandit algorithm. Several recent algorithms, different from GP-UCB and GP-TS, have been developed to alleviate the suboptimal and superlinear regret bounds in kernelized bandits and obtain an $\tilde{\mathcal{O}}(\sqrt{\Gamma(T)T})$ regret bound (Li and Scarlett, 2022; Salgia et al., 2021), that matches the lower bound in the case of the Matérn family of kernels, up to logarithmic factors. The *Sup* variations of the UCB algorithms also obtain the optimal regret bound in the contextual kernel bandit setting with finite actions (Valko et al., 2013).

**Main contribution.** The RL setting presents a greater level of complexity compared to the bandit setting due to the Markovian dynamics. None of the solutions in Li and Scarlett (2022); Salgia et al. (2021); Valko et al. (2013) seem appropriate in the presence of MDP dynamics, thereby leaving the question of order optimal regret bounds largely open. In this work, we leverage the scaling of the kernel spectrum with the size of the domain to improve the regret bounds. We consider kernels with polynomial eigendecay on a hypercubical domain with side length $\rho$, where eigenvalues scale with $\rho^\alpha$ for some $\alpha > 0$. See Definition 1. This encompasses a large class of common kernels, including the Matérn family, for which, $\alpha = 2\nu$. The hypercube domain assumption is a technical formality

---

[2]Also, see the extended version on *arXiv* (Yang et al., 2020b).

that can be relaxed to other regular compact subsets of $\mathbb{R}^d$. In Section 3, we propose a domain partitioning kernel ridge regression based least-squares value iteration policy ($\pi$-KRVI) that achieves sublinear regret of $\tilde{\mathcal{O}}(H^2 T^{\frac{d+\alpha/2}{d+\alpha}})$ for kernels introduced in Definition 1. This is the first sublinear regret bound under such a general stetting. Moreover, with Matérn kernels, our regret bound matches the $\Omega(T^{\frac{\nu+d}{2\nu+d}})$ lower bound reported in in Scarlett et al. (2017) for the special case of kernelized bandits, up to a logarithmic factor.

Our proposed policy, $\pi$-KRVI, is based on least-squares value iteration (similar to KOVI, Yang et al. (2020a)). However, in order to effectively utilize the confidence intervals from kernel ridge regression, $\pi$-KRVI creates a partitioning of the state-action domain and builds the confidence intervals only based on the observations within the same partition element. The domain partitioning allows us to leverage the scaling of the kernel eigenvalues with respect to the domain size. The inspiration for this idea is drawn from $\pi$-GP-UCB algorithm introduced in Janz et al. (2020) for kernelized bandits. In comparison to Janz et al. (2020), $\pi$-KRVI and its analysis present greater complexity due to the Markovian dynamics in the MDP setting. Furthermore, we provide a finer analysis that significantly improves the results compared to Janz et al. (2020). Although Janz et al. (2020) obtained sublinear regret guarantees of $\tilde{\mathcal{O}}(T^{\frac{2\nu+d(2d+3)}{4\nu+d(2d+4)}})$ in the kernelized bandit setting with Matérn kernel, there still remained a polynomial in $T$ gap between their regret bounds and the lower bound reported in Scarlett et al. (2017). As a consequence of our results, we also close this gap.

There are several novel contributions in our analysis that lead to the improved and order optimal regret bounds. We establish confidence intervals for kernel ridge regression that apply uniformly to all functions in the state-action value function class (Theorem 1). A similar confidence interval was given in Yang et al. (2020a). We however provide flexibility with respect to setting the parameters of the confidence interval, that eventually contributes to the improved regret bounds, with a proper choice of parameters. We also derive bounds on the maximum information gain (Lemma 2) and the function class covering number (Lemma 3), taking into consideration the size of the state-action domain. These bounds are important for the analysis of our domain partitioning policy which effectively controls the number of observations utilized in kernel ridge regression by partitioning the domain into subdomains of diminishing size. These intermediate results may also be of general interest in similar problems.

The $\pi$-KRVI policy enjoys an efficient runtime, polynomial in $T$, and linear in $|\mathcal{A}|$, similar to the runtime of KOVI (Yang et al., 2020a). The dependency of the runtime on $|\mathcal{A}|$ limits the scope of the policy to finite $\mathcal{A}$, while allowing a continuous $\mathcal{S}$ (with $|\mathcal{S}|$ infinite). The assumption of finite $\mathcal{A}$ can be relaxed, provided there is an efficient optimizer of a certain state-action value function. See the details in Section 3.2.

**Other related work.** There is an extensive literature on the analysis of RL policies which do not rely on a generative model or an exploratory behavioral policy. The literature has primarily focused on the tabular setting (Jin et al., 2018; Auer et al., 2008; Bartlett and Tewari, 2012). The domain of potential applications for this setting is very limited, as in many real world problems, the state-action space is very large or even infinite. In response to this, recent literature has placed a notable emphasis on employing function approximation in RL, particularly within the context of generalized linear settings. This approach involves representing the value function or transition model through a linear transformation to a well-defined feature mapping. Important contributions include the work of Jin et al. (2020); Yao et al. (2014), as well as subsequent studies by Russo (2019); Zanette et al. (2020a,b); Neu and Pike-Burke (2020); Yang and Wang (2020). Furthermore, there have been several efforts to extend these techniques to a kernelized setting, as explored in Yang et al. (2020a); Yang and Wang (2020); Chowdhury and Gopalan (2019); Yang et al. (2020c); Domingues et al. (2021). These works are also inspired by methods originally designed for linear bandits (Abbasi-Yadkori et al., 2011; Agrawal and Goyal, 2013), as well as kernelized bandits (Srinivas et al., 2010b; Valko et al., 2013; Chowdhury and Gopalan, 2017). However, all known regret bounds in the RL setting (Yang et al., 2020a; Yang and Wang, 2020; Chowdhury and Gopalan, 2019; Yang et al., 2020c; Domingues et al., 2021) are not order optimal. We compare our regret bounds with the state of the art reported in Yang et al. (2020a). A similar issue existed for classic kernelized bandit algorithms. A detailed discussion can be found in Vakili et al. (2021d). The authors in Yang and Wang (2020) considered finite state-actions under a kernelized MDP model where the transition model can be directly estimated. That is different from the setting considered in our work and Yang et al. (2020a).

## 2  Preliminaries and Problem Formulation

In this section, we overview the background on episodic MDPs and kernel ridge regression.

### 2.1  Episodic Markov Decision Processes

An episodic MDP can be described by the tuple $M = (\mathcal{S}, \mathcal{A}, H, P, r)$, where $\mathcal{S}$ is the state space, $\mathcal{A}$ is the action space, the integer $H$ is the length of each episode, $r = \{r_h\}_{h=1}^{H}$ are the reward functions and $P = \{P_h\}_{h=1}^{H}$ are the transition probability distributions.[2] We use the notation $\mathcal{Z} = \mathcal{S} \times \mathcal{A}$ to denote the state-action space. For each $h \in [H]$, the reward $r_h : \mathcal{Z} \to [0, 1]$ is the reward function at step $h$, which is supposed to be deterministic for simplicity, and $P_h(\cdot|s, a)$ is the transition probability distribution on $\mathcal{S}$ for the next state from state-action pair $(s, a)$. The choice of deterministic rewards allows us to concentrate on the core complexities of the problem, and should not be regarded as a limitation. Both the framework and results can be readily extended to a setting with random rewards.

A policy $\pi = \{\pi_h\}_{h=1}^{H}$, at each step $h$, determines the (possibly random) action $\pi_h : \mathcal{S} \to \mathcal{A}$ taken by the agent at state $s$. At the beginning of each episode $t = 1, 2, \cdots$, the environment picks an arbitrary state $s_1^t$. The agent determines a policy $\pi^t = \{\pi_h^t\}_{h=1}^{H}$. Then, at each step $h \in [H]$, the agent observes the state $s_h^t \in \mathcal{S}$, picks an action $a_h^t = \pi_h^t(s_h^t)$ and observes the reward $r_h(s_h^t, a_h^t)$. The new state $s_{h+1}^t$ then is drawn from the transition distribution $P_h(\cdot|s_h^t, a_h^t)$. The episode ends when the agent receives the final reward $r_H(s_H^t, a_H^t)$.

The goal is to find a policy $\pi$ that maximizes the expected total reward in the episode, starting at step $h$, i.e., the value function defined as

$$V_h^{\pi}(s) = \mathbb{E}\left[\sum_{h'=h}^{H} r_{h'}(s_{h'}, a_{h'}) \Big| s_h = s\right], \quad \forall s \in \mathcal{S}, h \in [H], \tag{2}$$

where the expectation is taken with respect to the randomness in the trajectory $\{(s_h, a_h)\}_{h=1}^{H}$ obtained by the policy $\pi$. It can be shown that under mild assumptions (e.g., continuity of $P_h$, compactness of $\mathcal{Z}$, and boundedness of $r$) there exists an optimal policy $\pi^{\star}$ which attains the maximum possible value of $V_h^{\pi}(s)$ at every step and at every state (e.g., see, Puterman, 2014). We use the notation $V_h^{\star}(s) = \max_{\pi} V_h^{\pi}(s)$, $\forall s \in \mathcal{S}, h \in [H]$. By definition $V_h^{\pi^{\star}} = V_h^{\star}$. For a value function $V : \mathcal{S} \to [0, H]$, we define the following notation

$$[P_h V](s, a) := \mathbb{E}_{s' \sim P_h(\cdot|s, a)}[V(s')]. \tag{3}$$

We also define the state-action value function $Q_h^{\pi} : \mathcal{Z} \to [0, H]$ as follows.

$$Q_h^{\pi}(s, a) = \mathbb{E}_{\pi}\left[\sum_{h'=h}^{H} r_{h'}(s_{h'}, a_{h'}) \Big| s_h = s, a_h = a\right], \tag{4}$$

where the expectation is taken with respect to the randomness in the trajectory $\{(s_h, a_h)\}_{h=1}^{H}$ obtained by the policy $\pi$. The Bellman equation associated with a policy $\pi$ then is represented as

$$Q_h^{\pi}(s, a) = r_h(s, a) + [P_h V_{h+1}^{\pi}](s, a), \quad V_h^{\pi}(s) = \mathbb{E}_{\pi}[Q_h^{\pi}(s, \pi_h(s))], \quad V_{H+1}^{\pi} := 0, \tag{5}$$

where the expectation is taken with respect to the randomness in the policy $\pi$. The Bellman optimality equation is also given as $Q_h^{\star}(s, a) = r_h(s, a) + [P_h V_{h+1}^{\star}](s, a)$, $V_h^{\star}(s) = \max_a Q_h^{\star}(s, a)$, $V_{H+1}^{\star} := 0$. The performance of a policy $\pi^t$ is measured in terms of the loss in the value function, referred to as *regret*, denoted by $\mathcal{R}(T)$ in the following definition

$$\mathcal{R}(T) = \sum_{t=1}^{T} (V_1^{\star}(s_1^t) - V_1^{\pi^t}(s_1^t)). \tag{6}$$

Recall that $\pi^t$ is the policy executed by the agent at episode $t$, where $s_1^t$ is the initial state in that episode determined by the environment.

---

[2]We intentionally do note use the standard term transition kernel for $P_h$, to avoid confusion with the term kernel in kernel-based learning.

## 2.2 Kernel Ridge Regression

We assume that the state-action value functions belong to a known reproducing kernel Hilbert space (RKHS). See Assumption 1 and Lemma 1 for the formal statement. This is a very general assumption, considering that the RKHS of common kernels can approximate almost all continuous functions on the compact subsets of $\mathbb{R}^d$ (Srinivas et al., 2010a). Consider a positive definite kernel $k : \mathcal{Z} \times \mathcal{Z} \to \mathbb{R}$. Let $\mathcal{H}_k$ be the RKHS induced by $k$, where $\mathcal{H}_k$ contains a family of functions defined on $\mathcal{Z}$. Let $\langle \cdot, \cdot \rangle_{\mathcal{H}_k} : \mathcal{H}_k \times \mathcal{H}_k \to \mathbb{R}$ and $\| \cdot \|_{\mathcal{H}_k} : \mathcal{H}_k \to \mathbb{R}$ denote the inner product and the norm of $\mathcal{H}_k$, respectively. The reproducing property implies that for all $f \in \mathcal{H}_k$, and $z \in \mathcal{Z}$, $\langle f, K(\cdot, z) \rangle_{\mathcal{H}_k} = f(z)$. Without loss of generality, we assume $k(z, z) \le 1$ for all $z$. Mercer theorem implies, under certain mild conditions, $k$ can be represented using an infinite dimensional feature map:

$$k(z, z') = \sum_{m=1}^{\infty} \sigma_m \phi_m(z) \phi_m(z'), \tag{7}$$

where $\sigma_m > 0$, and $\sqrt{\sigma_m} \phi_m \in \mathcal{H}_k$ form an orthonormal basis of $\mathcal{H}_k$. In particular, any $f \in \mathcal{H}_k$ can be represented using this basis and wights $w_m \in \mathbb{R}$ as

$$f = \sum_{m=1}^{\infty} w_m \sqrt{\sigma_m} \phi_m, \tag{8}$$

where $\|f\|_{\mathcal{H}_k}^2 = \sum_{m=1}^{\infty} w_m^2$. A formal statement and the details are provided in Appendix A. We refer to $\sigma_m$ and $\phi_m$ as (Mercer) eigenvalues and eigenfeatures of $k$, respectively.

Kernel-based models provide powerful predictors and uncertainty estimators which can be leveraged to guide the RL algorithm. In particular, consider a fixed unknown function $f \in \mathcal{H}_k$. Consider a set $Z^t = \{z^i\}_{i=1}^t \subset \mathcal{Z}$ of $t$ inputs. Assume $t$ noisy observations $\{Y(z^i) = f(z^i) + \varepsilon^i\}_{i=1}^t$ are provided, where $\varepsilon^i$ are independent zero mean noise terms. Kernel ridge regression provides the following predictor and uncertainty estimate, respectively (see, e.g., Schölkopf et al., 2002),

$$\begin{aligned} \mu^{t,f}(z) &= k_{Z^t}^\top(z)(K_{Z^t} + \lambda^2 I^t)^{-1} Y_{Z^t}, \\ (b^t(z))^2 &= k(z, z) - k_{Z^t}^\top(z)(K_{Z^t} + \lambda^2 I)^{-1} k_{Z^t}(z), \end{aligned} \tag{9}$$

where $k_{Z^t}(z) = [k(z, z^1), \dots, k(z, z^t)]^\top$ is a $t \times 1$ vector of the kernel values between $z$ and observations, $K_{Z^t} = [k(z^i, z^j)]_{i,j=1}^t$ is the $t \times t$ kernel matrix, $Y_{Z^t} = [Y(z^1), \dots, Y(Z^t)]^\top$ is the $t \times 1$ observation vector, $I$ is the identity matrix of dimensions $t$, and $\lambda > 0$ is a free regularization parameter. The predictor and uncertainty estimate could be interpreted as posterior mean and variance of a surrogate centered Gaussian process (GP) model with covariance $k$, and zero mean Gaussian noise with variance $\lambda^2$ (e.g., see, Williams and Rasmussen, 2006).

## 2.3 Technical Assumption

We assume that the reward functions $\{r_h\}_{h=1}^H$ and the transition probability distributions $P_h(s'|\cdot, \cdot)$ belong to the 1-ball of the RKHS. We use the notation $\mathcal{B}_{k,R} = \{f : \|f\|_{\mathcal{H}_k} \le R\}$ to denote the $R$-ball of the RKHS.

**Assumption 1** *We assume*

$$r_h(\cdot, \cdot), P_h(s'|\cdot, \cdot) \in \mathcal{B}_{k,1}, \quad \forall h \in [H], \forall s' \in \mathcal{S}. \tag{10}$$

This is a mild assumption considering the generality of RKHSs, that is also supposed to hold in Yang et al. (2020a). Similar assumptions are made in linear MDPs which are significantly more restrictive (e.g., see, Jin et al., 2020).

An immediate consequence of Assumption 1 is that for any integrable $V : \mathcal{S} \to [0, H]$, $r_h + [P_h V_{h+1}] \in \mathcal{B}_{k,H+1}$. This is formalized in the following lemma.

**Lemma 1** *Consider any integrable $V : \mathcal{S} \to [0, H]$. Under Assumption 1, we have*

$$r_h + [P_h V] \in \mathcal{B}_{k,H+1}. \tag{11}$$

See (Yeh et al., 2023, Lemma 3) for a proof.

# 3 Domain Partitioning Least-Squares Value Iteration Policy

A standard policy in episodic MDPs is the least-squares value iteration (LSVI), which computes an estimate $\widehat{Q}_h^t$ for $Q_h^\star$ at each step $h$ of episode $t$, by recursively applying Bellman equation as discussed in the previous section. In addition, an exploration bonus term $b_h^t : \mathcal{Z} \to \mathbb{R}$ is typically added leading to

$$Q_h^t = \min\{\widehat{Q}_h^t + \beta b_h^t, \; H - h + 1\}. \tag{12}$$

The term $\widehat{Q}_h^t + \beta b_h^t$ is an upper confidence bound on the state-action value function, that is inspired by the principle of *optimism in the face of uncertainty*. Since the rewards are assumed to be at most 1, the state-action value function at step $h$ is also bounded by $H - h + 1$. In episode $t$, then $\pi^t$ is the greedy policy with respect to $Q^t = \{Q_h^t\}_{h=1}^H$. Under Assumption 1, the estimate $\widehat{Q}_h^t$, the parameter $\beta$ and the exploration bonus $b_h^t$ can all be designed using kernel ridge regression. Specifically, having the Bellman equation in mind, $\widehat{Q}_h^t$ is the (kernel ridge) predictor for $r_h + [P_h V_{h+1}^t]$ using (possibly some of) the past $t - 1$ observations $\{r_h(z_h^\tau) + V_{h+1}^t(s_{h+1}^\tau)\}_{\tau=1}^{t-1}$ at points $\{z_h^\tau\}_{\tau=1}^{t-1}$. Recall that $\mathbb{E}\left[r_h(z_h^\tau) + V_{h+1}^t(s_{h+1}^\tau)\right] = r_h(z_h^\tau) + [P_h V_{h+1}^t](z_h^\tau)$, where the expectation is taken with respect to $P_h(\cdot|z_h^\tau)$. The observation noise $V_{h+1}^t(s_{h+1}^\tau) - [P_h V_{h+1}^t](z_h^\tau)$ is due to random transitions and is bounded by $H - h \leq H$.

## 3.1 Domain Partitioning

To overcome the suboptimal performance guarantees rooted in the online confidence intervals in kernel ridge regression, we introduce domain partitioning kernel ridge regression based least-squares value iteration ($\pi$-KRVI). The proposed policy partitions the state-action space $\mathcal{Z}$ into subdomains and builds kernel ridge regression only based on the observations within each subdomain. By doing so, we obtain tighter confidence intervals, ultimately resulting in a tighter regret bound. To formalize this procedure, we consider the state-action space $\mathcal{Z} \subset [0,1]^d$. Let $\mathcal{S}_h^t$, $h \in [H], t \in [T]$ be sets of hypercubes overlapping only at edges, covering the entire $[0,1]^d$. For any hypercube $\mathcal{Z}' \in \mathcal{S}_h^t$, we use $\rho_{\mathcal{Z}'}$ to denote the length of any of its sides, and $N_h^t(\mathcal{Z}')$ to denote the number of observations at step $h$ in $\mathcal{Z}'$ up to episode $t$:

$$N_h^t(\mathcal{Z}') = \sum_{\tau=1}^t \mathbf{1}\{(s_h^\tau, a_h^\tau) \in \mathcal{Z}'\}. \tag{13}$$

For all $h \in [H]$, we initialize $\mathcal{S}_h^1 = \{[0,1]^d\}$. At each episode $t$, for each step $h$, after observing a sample from $r_h + [P_h V_{h+1}^t]$ at $(s_h^t, a_h^t)$, we construct a new cover $\mathcal{S}_h^t$ as follows. We divide every element $\mathcal{Z}' \in \mathcal{S}_h^{t-1}$ that satisfies $\rho_{\mathcal{Z}'}^{-\alpha} < |N_h^t(\mathcal{Z}')| + 1$, into two equal halves along each side, generating $2^d$ hypercubes. The parameter $\alpha > 0$ in the splitting rule is a constant specified in Definition 1. Subsequently, we define $\mathcal{S}_h^t$ as the set of newly created hypercubes and the elements of $\mathcal{S}_h^{t-1}$ that were not split.

The construction of the cover sets described above ensures the number $N_h^t(\mathcal{Z}')$ of observations within each cover element $\mathcal{Z}'$ remains relatively small with respect to the size of $\mathcal{Z}'$, while also controlling the total number $|\mathcal{S}_h^t|$ of cover elements. The key parameter managing this tradeoff is $\alpha$, which is carefully chosen to achieve an appropriate width for the confidence interval, as shown in Section 4.

## 3.2 $\pi$-KRVI

Our proposed policy, $\pi$-KRVI, is derived by adopting the precise structure of an optimistic LSVI, as described previously, where the predictor and the exploration bonus are designed based on kernel ridge regression only on cover elements. In particular, for $z \in \mathcal{Z}$, let $\mathcal{Z}_h^t(z) \in \mathcal{S}_h^t$ be the cover element at step $h$ of episode $t$ containing $z$. Define $Z_h^t(z) = \{(s_h^\tau, a_h^\tau) \in \mathcal{Z}_h^t(z), \tau < t\}$ to be the set of past observations belonging to the same cover element as $z$. We then set

$$\widehat{Q}_h^t(z) = k_{Z_h^t(z)}^\top(z)(K_{Z_h^t(z)} + \lambda^2 I)^{-1} Y_{Z_h^t(z)}, \tag{14}$$

where $k_{Z_h^t(z)} = [k(z,z')]_{z' \in Z_h^t(z)}^\top$ is the kernel values between $z$ and all observations $z'$ in $Z_h^t(z)$, $K_{Z_h^t(z)} = [k(z',z'')]_{z',z'' \in Z_h^t(z)}$ is the kernel matrix for observations in $Z_h^t(z)$, and $Y_{Z_h^t(z)} =$

$[r_h(z') + V_{h+1}^t(s'_{h+1})]_{z' \in Z_h^t(z)}^\top$, where $s'_{h+1}$ is drawn from the transition distribution $P_h(\cdot|z')$, denotes the observation values for the observation points $z' \in Z_h^t(z)$. The vectors $k_{Z_h^t(z)}$ and $Y_{Z_h^t(z)}$ are $N_h^{t-1}(\mathcal{Z}_h^t(z))$ dimensional column vectors, and $K_{Z_h^t(z)}$ and $I$ are $N_h^{t-1}(\mathcal{Z}_h^t(z)) \times N_h^{t-1}(\mathcal{Z}_h^t(z))$ dimensional matrices.

The exploration bonus is determined based on the uncertainty estimate of the kernel ridge regression model on cover elements defined as

$$b_h^t(z) = \left( k(z,z) - k_{Z_h^t(z)}^\top(z)(K_{Z_h^t(z)} + \lambda^2 I)^{-1} k_{Z_h^t(z)}(z) \right)^{\frac{1}{2}}. \tag{15}$$

The policy $\pi$-KRVI then is the greedy policy with respect to

$$Q_h^t(z) = \min\{\widehat{Q}_h^t(z) + \beta_T(\delta)b_h^t(z), H - h + 1\}. \tag{16}$$

Specifically, at step $h$ of episode $t$, the following action is chosen, after observing $s_h^t$,

$$a_h^t = \arg\max_{a \in \mathcal{A}} Q_h^t(s_h^t, a). \tag{17}$$

A pseudocode is provided in Algorithm 1.

---

**Algorithm 1** The $\pi$-KRVI Policy.

---
1: Input: $\lambda$, $\beta_T(\delta)$, $k$, $M = (\mathcal{S}, \mathcal{A}, H, P, r)$.
2: For all $h \in [H]$, let $\mathcal{S}_h^1 = \{[0,1]^d\}$.
3: **for** Episode $t = 1, 2, \ldots, T$, **do**
4:     Receive the initial state $s_1^t$.
5:     Set $V_{H+1}^t(s) = 0$, for all $s$.
6:     **for** step $h = H, \ldots, 1$ **do**
7:         Obtain value functions $Q_h^t(z)$ as in (16).
8:     **end for**
9:     **for** step $h = 1, 2, \ldots, H$ **do**
10:         Take action $a_h^t \leftarrow \arg\max_{a \in \mathcal{A}} Q_h^t(s_h^t, a)$.
11:         Observe the reward $r_h(s_h^t, a_h^t)$ and the next state $s_{h+1}^t$.
12:         Split any element $\mathcal{Z}' \in \mathcal{S}_h^{t-1}$, for which $\rho_{\mathcal{Z}'}^{-\alpha} < |N_h^t(\mathcal{Z}')| + 1$ along the middle of each side, and obtain $\mathcal{S}_h^t$.
13:     **end for**
14: **end for**

---

The predictor $\widehat{Q}_h^t$, the confidence interval width multiplier $\beta_T(\delta)$ and the exploration bonus $b_h^t$ are all designed using kernel ridge regression limited to the observations within cover elements given above. The parameter $\beta_T(\delta)$, in particular, is designed in a way that $Q_h^t$ is a $1 - \delta$ upper confidence bound on $r_h + [P_h V_{h+1}^t]$. Using Theorem 1 on the confidence intervals, we show that a choice of $\beta_T(\delta) = \Theta(H\sqrt{\log(\frac{TH}{\delta})})$ satisfies this requirement.

Figure 1 demonstrates the domain partitioning used in $\pi$-KRVI on a 2-dimensional domain. The colors represent the value of the target function. The observation points are expected to concentrate around the areas where the target function has a high value. As a result the domain is partitioned to smaller squares in that region.

**Runtime complexity.** The $\pi$-KRVI policy is also runtime efficient with a polynomial runtime complexity. In particular, an upper bound on the runtime of $\pi$-KRVI is $\mathcal{O}(HT^4 + H|\mathcal{A}|T^3)$, that is similar to KOVI (Yang et al., 2020a). However, analogous to (Janz et al., 2020), we expect an improved runtime for $\pi$-KRVI in practice. In addition, the runtime can further improve in terms of $T$, utilizing sparse approximations of kernel ridge predictor and uncertainty estimate (e.g., see, Vakili et al., 2022). The dependency of the runtime on $|\mathcal{A}|$ is due to the step given in Equation (17). If this optimization can be done efficiently over continuous domains, $\pi$-KRVI (also KOVI) could handle infinite number of actions. The assumption that the upper confidence bound index can be efficiently optimized over continuous domains is often made in the kernelized bandits (e.g., see, Srinivas et al., 2010a).

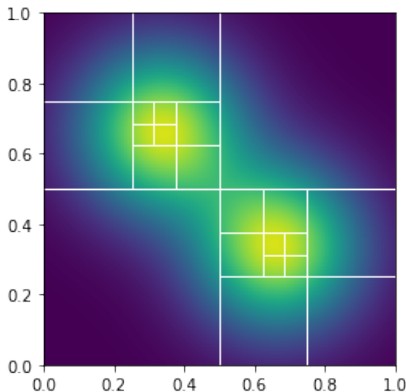

Figure 1: A 2-dimensional domain partitioned into smaller squares.

## 4 Main Results and Regret Analysis

In this section, we present our main results. In Theorem 2, we establish an $\tilde{\mathcal{O}}(H^2 T^{\frac{d+\alpha/2}{d+\alpha}})$ regret bound for $\pi$-KRVI, for the class of kernels with polynomial eigendecay. We first prove bounds on maximum information gain and covering number of state-action value function class. Those enable us to present our uniform confidence interval for state-action value functions (Theorem 1), and subsequently the regret bound (Theorem 2).

**Definition 1 (Polynomial Eigendecay)** *Consider the Mercer eigenvalues $\{\sigma_m\}_{m=1}^{\infty}$ of $k : \mathcal{Z} \times \mathcal{Z} \to \mathbb{R}$, given in Equation (7), in a decreasing order, as well as the corresponding eigenfeatures $\{\phi_m\}_{m=1}^{\infty}$. Assume $\mathcal{Z}$ is a d-dimensional hypercube with side length $\rho_{\mathcal{Z}}$. For some $C_p, \alpha > 0, p > 1$, the kernel $k$ is said to have a polynomial eigendecay, if for all $m \in \mathbb{N}$, $\sigma_m \leq C_p m^{-p} \rho_{\mathcal{Z}}^{\alpha}$. In addition, for some $\eta \geq 0$, $m^{-p\eta} \phi_m(z)$ is uniformly bounded over all $m$ and $z$. We use the notation $\tilde{p} = p(1 - 2\eta)$.*

The polynomial eigendecay profile encompasses a large class of common kernels, e.g., the Matérn family of kernels. For a Matérn kernel with smoothness parameter $\nu$, $p = \frac{2\nu+d}{d}$ and $\alpha = 2\nu$ (e.g., see, Janz et al., 2020). Another example is the NT kernel (Arora et al., 2019). It has been shown that the RKHS of the NT kernel, when the activations are $s - 1$ times differentiable, is equivalent to the RKHS of a Matérn kernel with smoothness $\nu = s - \frac{1}{2}$ (Vakili et al., 2021b). For instance, the RKHS of an NT kernel with ReLU activations is equivalent to the RKHS of a Matérn kernel with $\nu = \frac{1}{2}$ (also known as the Laplace kernel). In this case, $p = 1 + \frac{1}{d}$ and $\alpha = 1$. The hypercube domain assumption is a technical formality that can be relaxed to other regular compact subsets of $\mathbb{R}^d$. The uniform boundedness of $m^{-p\eta} \phi_m(z)$ for some $\eta > 0$, also holds for a broad class of kernels, including the Matérn family, as discussed in (Yang et al., 2020a). Several works including (Vakili et al., 2021b; Kassraie and Krause, 2022), have employed an averaging technique over subsets of eigenfeatures, demonstrating that, for the bounds on information gain, the effective value of $\eta$ can be considered as $0$ in the case of Matérn and NT kernels.

### 4.1 Confidence Intervals for State-Action Value Functions

Confidence intervals are an important building block in the design and analysis of bandit and RL algorithms. For a fixed function $f$ in the RKHS of a known kernel, $1 - \delta$ confidence intervals of the form $|f(z) - \mu^{t,f}(z)| \leq \beta(\delta) b^t(z)$ are established in several works (Srinivas et al., 2010a; Chowdhury and Gopalan, 2017; Abbasi-Yadkori, 2013; Vakili et al., 2021a) under various assumptions. In our setting of interest, however, these confidence intervals cannot be directly applied. This is due to the randomness of the target function itself. Specifically, in our case, the target function is $r_h + [P_h V_{h+1}^t]$, which is not a fixed function due to the temporal dependence within an episode. An argument based on the covering number of the state-action value function class was used in Yang et al. (2020a) to establish uniform confidence intervals over all $z \in \mathcal{Z}$ and all $f$ in a specific function class. In Theorem 1, we prove a different confidence interval that offers flexibility with respect to setting the parameters of the confidence interval. Our approach leads to a more refined confidence interval,

which, with a proper choice of parameters, contributes to the improved regret bound achieved by our policy.

We first give a formal definition of the two complexity terms: maximum information gain and the covering number of the state-action value function class, which appear in our confidence intervals.

**Definition 2 (Maximum Information Gain)** *In the kernel ridge regression setting described in Section 2.2, the following quantity is referred to as maximum information gain:* $\Gamma_{k,\lambda}(t) = \max_{Z^t \subset \mathcal{Z}} \frac{1}{2} \log det(I + \frac{1}{\lambda^2} K_{Z^t})$.

Upper bounds on maximum information gain based on the spectrum of the kernel are established in Janz et al. (2020); Srinivas et al. (2010a); Vakili et al. (2021c). Maximum information gain is closely related to the *effective* dimension of the kernel. While the feature representation of common kernels is infinite dimensional, with a finite observation set, only a finite number of features have a significant impact on kernel ridge regression, that is referred to as the effective dimension. It has been shown that information gain and effective dimension are the same up to logarithmic factors (Calandriello et al., 2019). This observation offers an intuitive understanding of information gain.

**State-action value function class:** Let us use $\mathcal{Q}_{k,h}(R, B)$ to denote the class of state-action value functions. In particular for a set of observations $Z$, let $b_h(z)$ be the uncertainty estimate obtained from kernel ridge regression as given in (9). We define

$$\mathcal{Q}_{k,h}(R, B) = \left\{ Q : Q(z) = \min\left\{Q_0(z) + \beta b_h(z), \ H - h + 1\right\}, \ \|Q_0\|_{\mathcal{H}_k} \le R, \beta \le B, |Z| \le T \right\}.$$
(18)

**Definition 3 (Covering Set and Number)** *Consider a function class $\mathcal{F}$. For $\epsilon > 0$, we define the minimum $\epsilon$-covering set $\mathcal{C}(\epsilon)$ as the smallest subset of $\mathcal{F}$ that covers it up to an $\epsilon$ error in $l_\infty$ norm. That is to say, for all $f \in \mathcal{F}$, there exists a $g \in \mathcal{C}(\epsilon)$, such that $\|f - g\|_{l_\infty} \le \epsilon$. We refer to the size of $\mathcal{C}(\epsilon)$ as the $\epsilon$-covering number.*

We use the notation $\mathcal{N}_{k,h}(\epsilon; R, B)$ to denote the $\epsilon$-covering number of $\mathcal{Q}_{k,h}(R, B)$, that appears in the confidence interval.

In Lemmas 2 and 3, we establish bounds on $\Gamma_{k,\lambda}(t)$ and $\mathcal{N}_{k,h}(\epsilon; R, B)$, respectively.

**Lemma 2 (Maximum information gain)** *Consider a positive definite kernel $k : \mathcal{Z} \times \mathcal{Z} \to \mathbb{R}$, with polynomial eigendecay on a hypercube with side length $\rho_{\mathcal{Z}}$. The maximum information gain given in Definition 2 satisfies*

$$\Gamma_{k,\lambda}(T) = \mathcal{O}\left(T^{\frac{1}{p}}(\log(T))^{1-\frac{1}{p}} \rho_{\mathcal{Z}}^{\frac{\alpha}{p}}\right).$$

**Lemma 3 (Covering Number of $\mathcal{Q}_{k,h}(R, B)$)** *Recall the class of state-action value functions $\mathcal{Q}_{k,h}(R, B)$, where $k : \mathcal{Z} \times \mathcal{Z} \to \mathbb{R}$ satisfies the polynomial eigendecay on a hypercube with side length $\rho_{\mathcal{Z}}$. We have*

$$\log \mathcal{N}_{k,h}(\epsilon; R, B) = \mathcal{O}\left(\left(\frac{R^2 \rho_{\mathcal{Z}}^\alpha}{\epsilon^2}\right)^{\frac{1}{p-1}} \left(1 + \log\left(\frac{R}{\epsilon}\right)\right) + \left(\frac{B^2 \rho_{\mathcal{Z}}^\alpha}{\epsilon^2}\right)^{\frac{2}{p-1}} \left(1 + \log\left(\frac{B}{\epsilon}\right)\right)\right).$$

Our bound on maximum information gain is stronger than the ones presented in Yang et al. (2020a); Janz et al. (2020); Srinivas et al. (2010a) and is similar to the one given in Vakili et al. (2021c), in terms of dependency on $T$. Our bound on function class covering number is similar to the one given in Yang et al. (2020a), in terms of dependency on $T$. Both Lemmas 2 and 3 given in this work are, however, novel in terms of dependency on the domain size $\rho_{\mathcal{Z}}$, and are required for the analysis of our domain partitioning algorithm.

We next present the confidence interval. Proofs are given in the appendix.

**Theorem 1 (Confidence Interval)** *Let $\widehat{Q}_h^t$ and $b_h^t$ denote the kernel ridge predictor and uncertainty estimate of $r_h + [P_h V_{h+1}^t]$, using $t$ observations $\{V_{h+1}^t(s_{h+1}^\tau)\}_{\tau=1}^t$ at $Z_h^t = \{z_h^\tau\}_{\tau=1}^t \subset \mathcal{Z}$, where $s_{h+1}^\tau$ is the next state drawn from $P_h(\cdot|z_h^\tau)$. Let $R_T = 2H\sqrt{\Gamma_{k,\lambda}(T)}$. For $\epsilon, \delta \in (0, 1)$, with probability, at least $1 - \delta$, we have, $\forall z \in \mathcal{Z}, h \in [H]$ and $t \in [T]$,*

$$|r_h(z) + [P_h V_{h+1}^t](z) - \widehat{Q}_h^t(z)| \le \beta_h^t(\delta, \epsilon) b_h^t(z) + \epsilon,$$

where $\beta_h^t(\delta, \epsilon)$ is set to any value satisfying

$$\beta_h^t(\delta, \epsilon) \geq H + 1 + \frac{H}{\sqrt{2}}\sqrt{\Gamma_{k,\lambda}(t) + \log \mathcal{N}_{k,h}(\epsilon; R_T, \beta_h^t(\delta, \epsilon)) + 1 + \log\left(\frac{TH}{\delta}\right)} + \frac{3\sqrt{t}\epsilon}{\lambda}. \quad (19)$$

## 4.2 Regret of $\pi$-KRVI

A key step in the analysis of $\pi$-KRVI is to apply the confidence interval in Theorem 1 to a subdomain $\mathcal{Z}' \in \mathcal{S}_h^t$. By design of the splitting rule, we can prove that the maximum information gain corresponding to $\mathcal{Z}'$ satisfies $\Gamma_{k,\lambda}(N_h^T(\mathcal{Z}')) = \mathcal{O}(\log(T))$. In addition, we choose $\epsilon = \frac{H\sqrt{\log(\frac{TH}{\delta})}}{\sqrt{N_h^t(\mathcal{Z}')}}$, when applying the confidence interval at step $h$ of episode $t$ on this subdomain. That ensures $\log \mathcal{N}_{k,h}(\epsilon; R_{N_h^T(\mathcal{Z}')}, \beta_h^t(\delta, \epsilon)) = \mathcal{O}(\log(T))$. From these, and by applying a probability union bound over all subdomains $\mathcal{Z}'$ created in $\pi$-KRVI, we can deduce that the choice of $\beta_T(\delta) = \Theta(H\sqrt{\log(\frac{TH}{\delta})})$ with a sufficiently large constant, satisfies the requirements for confidence interval widths based on Theorem 1. The details are provided in the proof of Theorem 2 in Appendix D. Then, using standard tools from the analysis of optimistic LSVI algorithms, we arrive at the following regret bound.

**Theorem 2 (Regret of $\pi$-KRVI)** *Consider the $\pi$-KRVI policy described in Section 3.2, with $\beta_T(\delta) = \Theta(H\sqrt{\log(\frac{TH}{\delta})})$ with a sufficiently large constant implied in the $\Theta$ notation. Under Assumption 1, for kernels given in Definition 1, with probability at least $1 - \delta$, the regret of $\pi$-KRVI satisfies*

$$\mathcal{R}(T) = \mathcal{O}\left(H^2 T^{\frac{d+\alpha/2}{d+\alpha}} \log(T)\sqrt{\log\left(\frac{H}{\delta}\right)}\right). \quad (20)$$

The regret bound of $\pi$-KRVI presented in Theorem 2 is sublinear in $T$ when $\alpha > 0$, in contrast to the state of the art regret bound in Yang et al. (2020a). The $\mathcal{O}$ notation used in the expression above hides constants that depend on $p, \alpha$ and $d$. See Appendix D for more details. When specialized to the Matérn family of kernels, replacing $p = \frac{2\nu+d}{d}$ and $\alpha = 2\nu$, the regret bound becomes

$$\mathcal{R}(T) = \mathcal{O}\left(H^2 T^{\frac{\nu+d}{2\nu+d}} \log(T)\sqrt{\log\left(\frac{H}{\delta}\right)}\right). \quad (21)$$

In terms of $T$ scaling, this matches the lower bound for the special case of kernelized bandits (Scarlett et al., 2017), up to a $\log(T)$ factor.

## 5 Conclusion

The analysis of RL algorithms has predominantly focused on simple settings such as tabular or linear MDPs. Several recent studies have considered more general models, including representing the state-action value functions using RKHSs. Notably, the work in Yang et al. (2020a) derives regret bounds for an optimistic LSVI policy. However, the regret bounds in Yang et al. (2020a) are sublinear only when the eigenvalues of the kernel decay rapidly. In this work, we leveraged a domain partitioning technique, a uniform confidence interval for state-action value functions, and bounds on complexity terms based on the domain size to propose $\pi$-KRVI, which attains a sublinear regret bound for a general class of kernels. Moreover, our regret bounds match the lower bound derived for Matérn kernels in the special case of kernelized bandits, up to logarithmic factors. It remains an open problem whether the suboptimal regret bounds in the case of standard optimistic LSVI policies (such as KOVI, Yang et al., 2020a) represent a fundamental shortcoming or an artifact of the proof.

## Funding Disclosure

This work was funded by MediaTek Research.

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

# A  Mercer Theorem and the RKHSs

Mercer theorem (Mercer, 1909) provides a representation of the kernel in terms of an infinite dimensional feature map (e.g., see, Christmann and Steinwart, 2008, Theorem 4.49). Let $\mathcal{Z}$ be a compact metric space and $\mu$ be a finite Borel measure on $\mathcal{Z}$ (we consider Lebesgue measure in a Euclidean space). Let $L^2_\mu(\mathcal{Z})$ be the set of square-integrable functions on $\mathcal{Z}$ with respect to $\mu$. We further say a kernel is square-integrable if

$$\int_{\mathcal{Z}} \int_{\mathcal{Z}} k^2(z, z')\, d\mu(z) d\mu(z') < \infty.$$

**Theorem 3** *(Mercer Theorem) Let $\mathcal{Z}$ be a compact metric space and $\mu$ be a finite Borel measure on $\mathcal{Z}$. Let $k$ be a continuous and square-integrable kernel, inducing an integral operator $T_k : L^2_\mu(\mathcal{Z}) \to L^2_\mu(\mathcal{Z})$ defined by*

$$(T_k f)(\cdot) = \int_{\mathcal{Z}} k(\cdot, z') f(z')\, d\mu(z'),$$

*where $f \in L^2_\mu(\mathcal{Z})$. Then, there exists a sequence of eigenvalue-eigenfeature pairs $\{(\sigma_m, \phi_m)\}_{m=1}^\infty$ such that $\sigma_m > 0$, and $T_k \phi_m = \sigma_m \phi_m$, for $m \geq 1$. Moreover, the kernel function can be represented as*

$$k(z, z') = \sum_{m=1}^\infty \sigma_m \phi_m(z) \phi_m(z'),$$

*where the convergence of the series holds uniformly on $\mathcal{Z} \times \mathcal{Z}$.*

According to the Mercer representation theorem (e.g., see, Christmann and Steinwart, 2008, Theorem 4.51), the RKHS induced by $k$ can consequently be represented in terms of $\{(\sigma_m, \phi_m)\}_{m=1}^\infty$.

**Theorem 4** *(Mercer Representation Theorem) Let $\{(\sigma_m, \phi_m)\}_{i=1}^\infty$ be the Mercer eigenvalue eigenfeature pairs. Then, the RKHS of $k$ is given by*

$$\mathcal{H}_k = \left\{ f(\cdot) = \sum_{m=1}^\infty w_m \sigma_m^{\frac{1}{2}} \phi_m(\cdot) : w_m \in \mathbb{R}, \|f\|_{\mathcal{H}_k}^2 := \sum_{m=1}^\infty w_m^2 < \infty \right\}.$$

Mercer representation theorem indicates that the scaled eigenfeatures $\{\sqrt{\sigma_m}\phi_m\}_{m=1}^\infty$ form an orthonormal basis for $\mathcal{H}_k$.

# B  Proof of Theorem 1 (Confidence Interval)

Confidence bounds of the form given in Theorem 1 have been established for a fixed function $f$ with bounded RKHS norm and sub-Gaussian observation noise in several works including Abbasi-Yadkori (2013); Chowdhury and Gopalan (2017); Vakili et al. (2021a). In the RL setting, however, we apply the confidence interval to $f = r_h + [P_h V_{h+1}^t]$. Although the RKHS norm of this target function is bounded by $H + 1$, this is not a fixed function as it depends on $V_{h+1}^t$. In addition the observation noise terms $V_{h+1}(s_{h+1}^t) - [P_h V_{h+1}^t](s_h^t, a_h^t)$ also depend on $V_{h+1}^t$. To handle this setting, we prove a confidence interval that holds for all possible $V_{h+1}^t : \mathcal{S} \to [0, H]$. For this purpose, we use a probability union bound and a covering set argument over the function class of $V_{h+1}^t$.

We first recall the confidence interval for a fixed function and noise sequence given in (Chowdhury and Gopalan, 2017, Theorem 2). See also (Abbasi-Yadkori, 2013, Corollary 3.15).

**Lemma 4** *Let $\{z^t \in \mathcal{Z}\}_{t=1}^T$ be a stochastic process predictable with respect to the filtration $\{\mathcal{F}_t\}_{t=0}^T$. Let $\{\varepsilon^t\}_{t=1}^T$ be a real valued $\mathcal{F}_t$ measurable stochastic process with a $\sigma$ sub-Gaussian distribution conditioned on $\mathcal{F}_{t-1}$. Let $\mu^{t,f}$ and $b^t$ be the kernel ridge predictor and uncertainty estimate of $f$ using $t$ noisy observations of the form $\{f(z^\tau) + \varepsilon^\tau\}_{\tau=1}^t$. Assume $f \in \mathcal{B}_{k,R}$. Then with probability at least $1 - \delta$, for all $z \in \mathcal{Z}$ and $t \geq 1$,*

$$|f(z) - \mu^{t,f}(z)| \leq \beta_1 b^t(z), \tag{22}$$

*where $\beta_1 = R + \sigma\sqrt{2(\Gamma_{k,\lambda}(t) + 1 + \log(\frac{1}{\delta}))}$.*

As discussed above, we cannot directly use this confidence interval on $r_h + [P_h V_{h+1}^t]$ in the RL setting. Instead, we need to prove a new confidence interval that holds true for all possible $V_{h+1}^t$. We thus define $\mathcal{V}$ to be the function class of $V_{h+1}^t$ as follows.

$$\mathcal{V}_{k,h}(R, B) = \{V : V(s) = \max_{a \in \mathcal{A}} Q(s, a), \text{ for some } Q \in \mathcal{Q}_{k,h}(R, B)\}. \tag{23}$$

For simplicity of presentation, we specify the parameters $R$ and $B$ later.

Let $\mathcal{C}_{k,h}^{\mathcal{V}}(\epsilon; R, B)$ be the smallest $\epsilon$-covering set of $\mathcal{V}_{k,h}(R, B)$ in terms of $l_\infty$ norm. That is to say for all $V \in \mathcal{V}_{k,h}(R, B)$, there exists some $\overline{V} \in \mathcal{C}_{k,h}^{\mathcal{V}}(\epsilon; R, B)$ such that $\|V - \overline{V}\|_{l_\infty} \leq \epsilon$. Let $\mathcal{N}_{k,h}^{\mathcal{V}}(\epsilon; R, B)$ denote the $\epsilon$ covering number of $\mathcal{V}_{k,h}(R, B)$. By definition $\mathcal{N}_{k,h}^{\mathcal{V}}(\epsilon; R, B) = |\mathcal{C}_{k,h}^{\mathcal{V}}(\epsilon; R, B)|$.

We can create a confidence bound for all $\overline{V} \in \mathcal{C}_{k,h}^{\mathcal{V}}(\epsilon; R, B)$, using Lemma 4 and a probability union bound over $\mathcal{C}_{k,h}^{\mathcal{V}}(\epsilon; R, B)$. Fix $h \in [H]$ and $t \in [T]$. Let us use the notation $\widehat{\overline{Q}}^t$ for the kernel ridge predictor with $\overline{V}$. That is $\widehat{\overline{Q}}^t(z) = k_{Z^t}^\top(z)(K_{Z^t} + \lambda^2 I)^{-1}\overline{Y}$, where $\overline{Y}^\top = [\overline{V}(s_{h+1}^\tau)]_{\tau=1}^t$, and $s_{h+1}^\tau$ is the next state drawn randomly from probability distribution $P_h(\cdot|z_h^\tau)$. In addition, to simplify the notation, we use $g = r_h + [P_h\overline{V}]$ and $\mu^{t,g} = \widehat{\overline{Q}}^t$. Also, let $b^t(z) = (k(z, z) - k_{Z^t}^\top(z)(K_{Z^t} + \lambda^2 I)^{-1}k_{Z^t}(z))^{\frac{1}{2}}$. Then, we have, with probability at least $1 - \delta$, for all $\overline{V} \in \mathcal{C}_{k,h}^{\mathcal{V}}(\epsilon; R, B)$ and for all $z \in \mathcal{Z}$,

$$|g(z) - \mu^{t,g}(z)| \leq \beta_2 b^t(z), \tag{24}$$

where $\beta_2 = H + 1 + \frac{H}{\sqrt{2}}\sqrt{\Gamma_{k,\lambda}(t) + \log \mathcal{N}_{k,h}^{\mathcal{V}}(\epsilon; R, B) + 1 + \log(\frac{1}{\delta})}$.

Confidence interval (24) is a direct application of Lemma 4 and using a probability union bound over all $\overline{V} \in \mathcal{C}_{k,h}^{\mathcal{V}}(\epsilon; R, B)$. Note that, $\|r_h + P_h\overline{V}\|_{\mathcal{H}_k} \leq H + 1$ (Lemma 1). In addition, $\overline{V}(s_{h+1}^\tau) - [P_h\overline{V}](z_h^\tau) \in [0, H]$ for all $h$ and $\tau$. A bounded random variable in $[0, H]$ is a $H/2$ sub-Gaussian random variable based on Hoeffding inequality (Hoeffding, 1994).

Now, we extend the uniform confidence interval over all $\overline{V} \in \mathcal{C}_{k,h}^{\mathcal{V}}(\epsilon; R, B)$ to a uniform confidence interval over all $V \in \mathcal{V}_{k,h}(R, B)$. For some $V \in \mathcal{V}_{k,h}(R, B)$, define $f = r_h + [P_h V]$ and $\mu^{t,f} = \widehat{Q}^t$, similar to $g$ and $\mu^{t,g}$. By definition of $\mathcal{C}_{k,h}^{\mathcal{V}}(\epsilon; R, B)$, there exists $\overline{V} \in \mathcal{C}_{k,h}^{\mathcal{V}}(\epsilon; R, B)$, such that $\|V - \overline{V}\|_{l_\infty} \leq \epsilon$. Thus, for all $z \in \mathcal{Z}$,

$$f(z) - g(z) = [PV](z) - [P\overline{V}](z) \leq \sup_{s \in \mathcal{S}} |V(s) - \overline{V}(s)| \leq \epsilon. \tag{25}$$

Therefore, with probability at least $1 - \delta$,

$$
\begin{aligned}
|f(z) - \mu^{t,f}(z)| &\leq |f(z) - g(z)| + |g(z) - \mu^{t,g}(z)| + |\mu^{t,g}(z) - \mu^{t,f}(z)| \\
&\leq \beta_2 b^t(z) + \epsilon + |\mu^{t,g}(z) - \mu^{t,f}(z)|. \tag{26}
\end{aligned}
$$

Next, we prove that $|\mu^{t,f}(z) - \mu^{t,g}(z)| \leq \frac{3\epsilon\sqrt{t}b^t(z)}{\lambda}$.

Let us further simplify the notation by introducing $\alpha_t(z) = (K_{Z_t} + \lambda^2 I)^{-1}k_{Z_t}(z)$, $F_t^\top = [f(z_h^\tau)]_{\tau=1}^t$, $E_t^\top = [\varepsilon^\tau = V(s_{h+1}^\tau) - [P_h V](z_h^\tau)]_{\tau=1}^t$, $G_t^\top = [g(z_h^\tau)]_{\tau=1}^t$, $\overline{E}_t^\top = [\bar{\varepsilon}^\tau = \overline{V}(s_{h+1}^\tau) - [P_h\overline{V}](z_h^\tau)]_{\tau=1}^t$ so that $\mu^{t,f}(z) = \alpha^\top(z)(F_t + E_t)$ and $\mu^{t,g}(z) = \alpha^\top(z)(G_t + \overline{E}_t)$.

As discussed earlier, the observation noise terms $\varepsilon^t$ also depend on $V$. We have, for all $t \geq 1$,

$$
\begin{aligned}
|\varepsilon^t - \bar{\varepsilon}^t| &= \left|V(s_{h+1}^\tau) - \overline{V}(s_{h+1}^\tau) - ([P_h V](z_h^\tau) - [P_h\overline{V}](z_h^\tau))\right| \\
&\leq 2\epsilon.
\end{aligned}
$$

Using the difference between $f$ and $g$, as well as the difference between noise terms, we have

$$
\begin{aligned}
|\mu^{t,f}(z) - \mu^{t,g}(z)| &= |\alpha_t^\top(z)(F_t + E_t) - \alpha^\top(z)(G_t + \overline{E}_t)| \\
&\leq \|\alpha_t(z)\| \|F_t - G_t + E_t - \overline{E}_t\| \\
&\leq 3\epsilon\sqrt{t}\|\alpha_t(z)\| \\
&\leq \frac{3\epsilon\sqrt{t}b^t(z)}{\lambda},
\end{aligned}
$$

where the last inequality follows from $\|\alpha_t(z)\| \leq \frac{b^t(z)}{\lambda}$ (e.g., see, Vakili et al., 2021a, Proposition 1).

The bound on $|\mu^{t,f}(z) - \mu^{t,g}(z)|$ combined with (26) proves that for a fixed $t \in [T]$, fixed $h \in [H]$, for all $z \in \mathcal{Z}$ and for all $V \in \mathcal{V}_{k,h}(R, B)$,

$$
|f(z) - \mu^{t,f}(z)| \leq \beta_3 b^t(z) + \epsilon,
$$

where

$$
\beta_3 = H + 1 + \frac{H}{\sqrt{2}}\sqrt{\Gamma_{k,\lambda}(t) + \log\mathcal{N}_{k,h}^{\mathcal{V}}(\epsilon; R, B) + 1 + \log(\frac{1}{\delta})} + \frac{3\sqrt{t}\epsilon}{\lambda}. \tag{27}
$$

The confidence interval holds uniformly for all $h \in [H]$ and $t \in [T]$ using a probability union bound, when $\beta_3$ is replaced with the following

$$
\beta_4 = H + 1 + \frac{H}{\sqrt{2}}\sqrt{\Gamma_{k,\lambda}(t) + \log\mathcal{N}_{k,h}^{\mathcal{V}}(\epsilon; R, B) + 1 + \log(\frac{HT}{\delta})} + \frac{3\epsilon\sqrt{t}}{\lambda}. \tag{28}
$$

To complete the proof, we bound $\mathcal{N}_{k,h}^{\mathcal{V}}(\epsilon; R, B)$ in terms of the specific parameters of the problem. Firstly, the $\epsilon$-covering number of $\mathcal{V}_{k,h}(R, B)$ is bounded by that of $\mathcal{Q}_{k,h}(R, B)$ (Yang et al., 2020a, proof of Lemma $D$.1). Recall the definition of $\mathcal{Q}_{k,h}(R, B)$ in (18). We note that $\|\widehat{Q}_h^t\|_{\mathcal{H}_k} \leq 2H\sqrt{\Gamma_{k,\lambda}(t)}$ (Yang et al., 2020a, Lemma $C$.1). Thus, the theorem follows with $\beta_h^t(\delta, \epsilon)$, where $\beta_h^t(\delta, \epsilon)$ is set to some value satisfying

$$
\beta_h^t(\delta, \epsilon) \geq H + 1 + \frac{H}{\sqrt{2}}\sqrt{\Gamma_{k,\lambda}(t) + \log\mathcal{N}_{k,h}(\epsilon; R_t, \beta_h^t(\delta, \epsilon)) + 1 + \log(\frac{HT}{\delta})} + \frac{3\epsilon\sqrt{t}}{\lambda}, \tag{29}
$$

with $R_t = 2H\sqrt{\Gamma_{k,\lambda}(t)}$. That completes the proof of Theorem 1.

## C  Proof of Lemmas 2 (Maximum Information Gain) and 3 (Covering Number).

We first introduce the projection of the RKHS on a lower dimensional RKHS that is used in the proof of both lemmas. We then present the proofs. Recall the Mercer theorem and the representation of kernel using Mercer eigenvalues and eigenfeatures. Using Mercer representation theorem, any $f \in \mathcal{B}_R$ can be written as

$$
f = \sum_{m=1}^{\infty} w_m \sqrt{\sigma_m} \phi_m, \tag{30}
$$

with $\sum_{m=1}^{\infty} w_m^2 \leq R^2$. For some $D \in \mathbb{N}$, let $\Pi_D[f]$ denote the projection of $f$ onto the $D$-dimensional RKHS corresponding to the first $D$ features with the largest eigenvalues

$$
\Pi_D[f] = \sum_{m=1}^{D} w_m \sqrt{\sigma_m} \phi_m. \tag{31}
$$

Let us use the notations $\boldsymbol{w}_D = [w_1, w_2, \cdots, w_D]^\top$ for the $D$-dimensional column vector of weights, $\boldsymbol{\phi}_D(z) = [\phi_1(z), \phi_2(z), \cdots, \phi_D(z)]^\top$ for the $D$-dimensional column vector of eigenfeatures, and

$\Sigma_D = \mathrm{diag}([\sigma_1, \sigma_2, \cdots, \sigma_D])$ for the diagonal matrix of eigenvalues with $[\sigma_1, \sigma_2, \cdots, \sigma_D]$ as the diagonal entries. We also use the notations

$$k^D(z, z') = \phi_D^\top(z)\Sigma_D\phi_D(z), \tag{32}$$

to denote the kernel corresponding to the $D$-dimensional RKHS, as well as $k^0(z, z') = k(z, z') - k^D(z, z')$.

## C.1 Proof of Lemma 2 on Maximum Information Gain

Recall the definition of $\Gamma_{k,\lambda}(t)$. We have

$$
\begin{aligned}
\frac{1}{2}\log\det\left(I + \frac{1}{\lambda^2}K_{Z^t}\right) &= \frac{1}{2}\log\det\left(I + \frac{1}{\lambda^2}(K_{Z^t}^D + K_{Z^t}^0)\right) \\
&= \underbrace{\frac{1}{2}\log\det\left(I + \frac{1}{\lambda^2}K_{Z^t}^D\right)}_{\text{Term }(i)} + \underbrace{\frac{1}{2}\log\det\left(I + \frac{1}{\lambda^2}(I + \frac{1}{\lambda^2}K_{Z^t}^D)^{-1}K_{Z^t}^0\right)}_{\text{Term }(ii)}.
\end{aligned}
$$

We next bound the two terms on the right hand side.

**Term $(i)$:** Note that for $k^D$ corresponding to the $D$-dimensional RKHS, we have $K_{Z^t}^D = \mathbf{\Phi}_t\Sigma_D\mathbf{\Phi}_t^\top$, where $\mathbf{\Phi}_t = [\phi_D(z)]_{z \in Z^t}^\top$ is a $t \times D$ matrix that stacks the feature vectors $\phi_D(z^\tau)$, $\tau = 1, \cdots, t$, as it rows. By Weinstein–Aronszajn identity (Pozrikidis, 2014) (a special case of matrix determinant lemma),

$$
\begin{aligned}
\log\det\left(I^t + \frac{1}{\lambda^2}K_{Z^t}^D\right) &= \log\det\left(I^t + \frac{1}{\lambda^2}\mathbf{\Phi}_t\Sigma_D\mathbf{\Phi}_t^\top\right) \tag{33} \\
&= \log\det\left(I^D + \frac{1}{\lambda^2}\Sigma_D^{\frac{1}{2}}\mathbf{\Phi}_t\mathbf{\Phi}_t^\top\Sigma_D^{\frac{1}{2}}\right) \\
&\leq D\log\left(\frac{\mathrm{tr}(I^D + \frac{1}{\lambda^2}\Sigma_D^{\frac{1}{2}}\mathbf{\Phi}_t\mathbf{\Phi}_t^\top\Sigma_D^{\frac{1}{2}})}{D}\right) \\
&\leq D\log\left(1 + \frac{t}{\lambda^2 D}\right).
\end{aligned}
$$

The first inequality follows from the inequality of arithmetic and geometric means on eigenvalues of the argument, and the second inequality follows from $k^D \leq 1$. For clarity, we used the notations $I^t$ and $I^D$ for identity matrices of dimension $t$ and $D$, respectively. Otherwise, we drop the superscript.

**Term $(ii)$:** Similarly using the inequality of arithmetic and geometric means on eigenvalues, we bound the log det by the log of the trace of the argument. Let us use $\epsilon_D$ to denote an upper bound on $k^0$.

$$
\begin{aligned}
\log\det\left(I + \frac{1}{\lambda^2}(I + \frac{1}{\lambda^2}K_{Z^t}^D)^{-1}K_{Z^t}^0\right) &\leq t\log\left(\frac{\mathrm{tr}(I + \frac{1}{\lambda^2}(I + \frac{1}{\lambda^2}K_{Z^t}^D)^{-1}K_{Z^t}^0)}{t}\right) \tag{34} \\
&\leq t\log\left(1 + \frac{\epsilon_D}{\lambda^2}\right) \\
&\leq \frac{t\epsilon_D}{\lambda^2}.
\end{aligned}
$$

The last inequality uses $\log(1 + x) \leq x$ which holds for all $x \in \mathbb{R}$.

Combining the bounds on Term $(i)$ and Term $(ii)$, we have

$$\Gamma_{k,\lambda}(t) \leq \frac{D}{2}\log\left(1 + \frac{t}{\lambda^2 D}\right) + \frac{t\epsilon_D}{2\lambda^2}. \tag{35}$$

Now, using the polynomial eigendecay profile given in Definition 2,

$$
\begin{align}
k^0(z, z') &= \sum_{m=D+1}^{\infty} \sigma_m \phi_m(z) \phi_m(z') \tag{36} \\
&\leq C_1^2 C_p \rho_{\mathcal{Z}}^{\alpha} \sum_{m=D+1}^{\infty} m^{-p(1-2\eta)} \\
&\leq C_1^2 C_p \rho_{\mathcal{Z}}^{\alpha} \int_D^{\infty} x^{-\tilde{p}} dx \\
&\leq \frac{C_1^2 C_p \rho_{\mathcal{Z}}^{\alpha}}{\tilde{p}-1} D^{-\tilde{p}+1}. \tag{37}
\end{align}
$$

The constant $C_1$ is the uniform bound on $m^{-p\eta}\phi_m$, and $C_p$ is the parameter in Definition 1.

Choosing $D = Ct^{\frac{1}{\tilde{p}}} \rho_{\mathcal{Z}}^{\frac{\alpha}{\tilde{p}}} (\log(t))^{-\frac{1}{\tilde{p}}}$, with constant $C = (\frac{C_1^2 C_p}{(\tilde{p}-1)\lambda^2})^{\frac{1}{\tilde{p}}}$ we obtain

$$
\Gamma_{k,\lambda}(t) \leq \frac{C}{2} t^{\frac{1}{\tilde{p}}} \rho_{\mathcal{Z}}^{\frac{\alpha}{\tilde{p}}} \left( \log(t)^{-\frac{1}{\tilde{p}}} \log(1 + \frac{t}{\lambda^2 D}) + (\log(t))^{1-\frac{1}{\tilde{p}}} \right), \tag{38}
$$

that completes the proof.

## C.2 Proof of Lemma 3 on Covering Number of State-Action Value Function Class

Recall the definition of the state-action value function class,

$$
\mathcal{Q}_{k,h}(R, B) = \left\{ Q : Q(z) = \min \left\{ Q_0(z) + \beta b(z), H - h + 1 \right\}, \|Q_0\|_{\mathcal{H}_k} \leq R, \beta \leq B, |Z| \leq T \right\}.
$$

and the notation $\mathcal{N}_{k,h}(\epsilon; R, B)$ for its $\epsilon$-covering number. Let us use the notation $\mathcal{N}_{k,R}(\epsilon)$ for the $\epsilon$-covering number of RKHS ball $\mathcal{B}_{k,R} = \{ f : \|f\|_{\mathcal{H}_k} \leq R \}$, $\mathcal{N}_{[0,B]}(\epsilon)$ for the $\epsilon$-covering number of interval $[0, B]$ with respect to Euclidean distance, and $\mathcal{N}_{k,\boldsymbol{b}}(\epsilon)$ for the $\epsilon$-covering number of class of uncertainty functions $\boldsymbol{b}_k = \{ b(z) = \left( k(z, z) - k_Z^{\top}(z)(K_Z + \lambda^2 I)^{-1} k_Z(z) \right)^{\frac{1}{2}}, |Z| \leq T \}$.

Consider $Q, \overline{Q} \in \mathcal{Q}_{k,h}(R, B)$ where $Q(z) = \min \{ Q_0(z) + \beta b(z), H - h + 1 \}$ and $\overline{Q}(z) = \min \{ \overline{Q}_0(z) + \bar{\beta}\bar{b}(z), H - h + 1 \}$. We have

$$
|Q(z) - \overline{Q}(z)| \leq |Q_0(z) - \overline{Q}_0(z)| + |\beta - \bar{\beta}| + B|b(z) - \bar{b}(z)|. \tag{39}
$$

That implies

$$
\mathcal{N}_{k,h}(\epsilon; R, B) \leq \mathcal{N}_{k,R}(\frac{\epsilon}{3}) \mathcal{N}_{[0,B]}(\frac{\epsilon}{3}) \mathcal{N}_{k,\boldsymbol{b}}(\frac{\epsilon}{3B}). \tag{40}
$$

For the $\epsilon$-covering number of the $[0, B]$ interval, we simply have $\mathcal{N}_{[0,B]}(\epsilon/3) \leq 1 + 3B/\epsilon$. In the next lemmas, we bound the $\epsilon$-covering number of the RKHS ball and the class of uncertainty functions.

**Lemma 5 (RKHS Covering Number)** *Consider a positive definite kernel $k : \mathcal{Z} \times \mathcal{Z} \to \mathbb{R}$, with polynomial eigendecay on a hypercube with side length $\rho_{\mathcal{Z}}$. The $\epsilon$-covering number of $R$-ball in the RKHS satisfies*

$$
\log \mathcal{N}_{k,R}(\epsilon) = \mathcal{O} \left( \left( \frac{R^2 \rho_{\mathcal{Z}}^{\alpha}}{\epsilon^2} \right)^{\frac{1}{\tilde{p}-1}} \log(1 + \frac{R}{\epsilon}) \right). \tag{41}
$$

**Lemma 6 (Uncertainty Class Covering Number)** *Consider a positive definite kernel $k : \mathcal{Z} \times \mathcal{Z} \to \mathbb{R}$, with polynomial eigendecay on a hypercube with side length $\rho_{\mathcal{Z}}$. The $\epsilon$-covering number of the class of uncertainty functions satisfies*

$$
\log \mathcal{N}_{k,\boldsymbol{b}}(\epsilon) = \mathcal{O} \left( (\frac{\rho_{\mathcal{Z}}^{\alpha}}{\epsilon^2})^{\frac{2}{\tilde{p}-1}} (1 + \log(\frac{1}{\epsilon})) \right) \tag{42}
$$

Combining (40) with Lemmas 5 and 6, we obtain

$$\log \mathcal{N}_{k,h}(\epsilon; R, B) = \mathcal{O}\left( (\frac{R^2 \rho_{\mathcal{Z}}^{\alpha}}{\epsilon^2})^{\frac{1}{\tilde{p}-1}}(1+\log(\frac{R}{\epsilon})) + (\frac{B^2 \rho_{\mathcal{Z}}^{\alpha}}{\epsilon^2})^{\frac{2}{\tilde{p}-1}}(1+\log(\frac{B}{\epsilon})) \right), \qquad (43)$$

that completes the proof of Lemma 3. Next, we provide the proof of two lemmas above on the covering numbers of the RKHS ball and the uncertainty function class.

**Proof 1 (Proof of Lemma 5)** *For $f$ in the RKHS, recall the following representation*

$$f = \sum_{m=1}^{\infty} w_m \sqrt{\sigma_m} \phi_m, \qquad (44)$$

*as well as its projection on the $D$-dimensional RKHS*

$$\Pi_D[f] = \sum_{m=1}^{D} w_m \sqrt{\sigma_m} \phi_m. \qquad (45)$$

*We note that*

$$
\begin{aligned}
\|f - \Pi_D[f]\|_\infty &= \sum_{m=D+1}^{\infty} w_m \sqrt{\sigma_m} \phi_m \\
&\leq C_1 C_p^{\frac{1}{2}} \rho_{\mathcal{Z}}^{\alpha/2} \sum_{m=D+1}^{\infty} |w_m| m^{-p(\frac{1}{2}-\eta)} \\
&\leq C_1 C_p^{\frac{1}{2}} \rho_{\mathcal{Z}}^{\alpha/2} \left( \sum_{m=D+1}^{\infty} |w_m|^2 \right)^{\frac{1}{2}} \left( \sum_{m=D+1}^{\infty} m^{-p(1-2\eta)} \right)^{\frac{1}{2}} \\
&\leq C_1 C_p^{\frac{1}{2}} \rho_{\mathcal{Z}}^{\alpha/2} R \left( \int_D^{\infty} x^{-\tilde{p}} dx \right)^{\frac{1}{2}} \\
&= \frac{C_1 C_p^{\frac{1}{2}} \rho_{\mathcal{Z}}^{\alpha/2} R}{\sqrt{\tilde{p}-1}} D^{\frac{-\tilde{p}+1}{2}}.
\end{aligned}
$$

*In the expressions above, $C_1$ is the uniform bound on $m^{-p\eta}\phi_m$, and $C_p$ is the constant specified in Definition 1. The third inequality follows form Cauchy–Schwarz inequality.*

*Now, let $D_0$ be the smallest $D$ such that the right hand side is bounded by $\frac{\epsilon}{2}$. There exists a constant $C_2 > 0$, only depending on constants $C_1$, $C_p$, $\eta$ and $\tilde{p}$, such that*

$$D_0 \leq C_2 \left( \frac{R^2 \rho_{\mathcal{Z}}^{\alpha}}{\epsilon^2} \right)^{\frac{1}{\tilde{p}-1}}. \qquad (46)$$

*For a $D$-dimensional linear model, where the norm of the weights is bounded by $R$, the $\epsilon$-covering is at most $C_3 D (1 + \log(\frac{R}{\epsilon}))$, for some constant $C_3$ (e.g., see, Yang et al., 2020a). Using an $\epsilon/2$ covering number for the space of $\Pi_D[f]$ and using the minimum number of dimensions that ensures $|f - \Pi_D[f]| \leq \epsilon/2$, we conclude that*

$$
\begin{aligned}
\log \mathcal{N}_{k,R}(\epsilon) &\leq C_3 D_0 (1 + \log(\frac{R}{\epsilon})) \\
&\leq C_2 C_3 \left( \frac{R^2 \rho_{\mathcal{Z}}^{\alpha}}{\epsilon^2} \right)^{\frac{1}{\tilde{p}-1}} (1 + \log(\frac{R}{\epsilon})),
\end{aligned}
$$

*that completes the proof of the lemma.*

**Proof 2 (Proof of Lemma 6)** *Let us define $\boldsymbol{b}_k^2 = \{b^2 : b \in \boldsymbol{b}_k\}$ and $\mathcal{N}_{k,\boldsymbol{b}^2}(\epsilon)$ to be its $\epsilon$-covering number. We note that, for $b, \bar{b} \in \boldsymbol{b}$,*

$$|b(z) - \bar{b}(z)| \leq \sqrt{|(b(z))^2 - (\bar{b}(z))^2|}. \tag{47}$$

*Thus, an $\epsilon$-covering number of $\boldsymbol{b}$ is bounded by an $\epsilon^2$-covering of $\boldsymbol{b}^2$:*

$$\mathcal{N}_{k,\boldsymbol{b}}(\epsilon) \leq \mathcal{N}_{k,\boldsymbol{b}^2}(\epsilon^2). \tag{48}$$

*We next bound $\mathcal{N}_{k,\boldsymbol{b}^2}(\epsilon^2)$.*

*Using the feature space representation of the kernel, we obtain*

$$(b(z))^2 = \sum_{m=1}^{\infty} \gamma_m \sigma_m \phi_m^2(z), \tag{49}$$

*for some $\gamma_m \in [0,1]$. Based on the GP interpretation of the model, $\gamma_m$ can be understood as the posterior variances of the weights. Let $D_0$ be the smallest $D$ such that $\sum_{m=D+1}^{\infty} \sigma_m \phi_m^2(z) \leq \epsilon^2/2$. From Equation (37), we can see that, for some constant $C_4$, only depending on constants $C_1, C_p, \eta$ and $\tilde{p}$,*

$$D_0 \leq C_4 \left( \frac{\rho_{\mathcal{Z}}^{\alpha}}{\epsilon^2} \right)^{\frac{1}{\tilde{p}-1}}. \tag{50}$$

*For $\sum_{m=1}^{D_0} \gamma_m \sigma_m \phi_m^2(z)$ on a finite $D_0$-dimensional spectrum, as shown in Lemma D.3 of Yang et al. (2020a), an $\epsilon^2/2$ covering number scales with $D_0^2$. Specifically, an $\epsilon^2/2$ covering number of the space of $\sum_{m=1}^{D_0} \gamma_m \sigma_m \phi_m^2(z)$ is bounded by*

$$C_5 D_0^2 (1 + \log(\frac{1}{\epsilon})). \tag{51}$$

*Combining Equations (50) and (51), we obtain*

$$\begin{aligned}
\mathcal{N}_{k,\boldsymbol{b}^2}(\epsilon^2) &\leq C_5 D_0^2 (1 + \log(\frac{1}{\epsilon})) \\
&\leq C_5 C_4^2 \left( \frac{\rho_{\mathcal{Z}}^{\alpha}}{\epsilon^2} \right)^{\frac{2}{\tilde{p}-1}},
\end{aligned}$$

*that completes the proof of the lemma.*

## D  Proof of Theorem 2 (Regret of $\pi$-KRVI).

Following the standard analysis of optimisitc LSVI policies, for any $h \in [H]$, $t \in [T]$, we define temporal difference error $\delta_h^t : \mathcal{Z} \to \mathbb{R}$ as

$$\delta_h^t(z) = r_h(z) + [P_h V_{h+1}^t](z) - Q_h^t(z), \quad \forall z \in \mathcal{Z}. \tag{52}$$

Roughly speaking, $\{\delta_h^t(z)\}_{h=1}^{H}$ quantify how far the $\{Q_h^t\}_{h=1}^{H}$ are from satisfying the Bellman optimality equation.

For any $h \in [H], t \in [T]$, we also define

$$\begin{aligned}
\xi_h^t &= \left( V_h^t(s_h^t) - V_h^{\pi^t}(s_h^t) \right) - \left( Q_h^t(z_h^t) - Q_h^{\pi^t}(z_h^t) \right), \\
\zeta_h^t &= \left( [P_h V_{h+1}^t](z_h^t) - [P_h V_{h+1}^{\pi^t}](z_h^t) \right) - \left( V_{h+1}^t(s_{h+1}^t) - V_{h+1}^{\pi^t}(s_{h+1}^t) \right).
\end{aligned} \tag{53}$$

Using the notation defined above, we then have the following regret decomposition into three parts.

**Lemma 7 (Lemma** 5.1 **in** Yang et al. (2020a) **on regret decomposition)** *We have*

$$\mathcal{R}(T) = \underbrace{\sum_{t=1}^{T}\sum_{h=1}^{H}\mathbb{E}_{\pi^{\star}}[\delta_h^t(z_h)|s_1 = s_1^t] - \delta_h^t(z_h^t)}_{(i)} + \underbrace{\sum_{t=1}^{T}\sum_{h=1}^{H}(\xi_h^t + \zeta_h^t)}_{(ii)}$$

$$+ \underbrace{\sum_{t=1}^{T}\sum_{h=1}^{H}\mathbb{E}_{\pi^{\star}}[Q_h^t(s_h, \pi_h^{\star}(s_h)) - Q_h^t(s_h, \pi_h^t(s_h))|s_1 = s_1^t]}_{(iii)}. \tag{54}$$

The third term is negative, by definition of $\pi_h^t$ that is the greedy policy with respect to $Q_h^t$:

$$Q_h^t(s_h, \pi_h^{\star}(s_h)) - Q_h^t(s_h, \pi_h^t(s_h)) = Q_h^t(s_h, \pi_h^{\star}(s_h)) - \max_{a \in \mathcal{A}} Q_h^t(s_h, a) \le 0,$$

for all $s_h \in \mathcal{S}$. The second term is bounded using the following lemma.

**Lemma 8 (Lemma** 5.3 **in** Yang et al. (2020a)**)** *For any $\delta \in (0, 1)$, with probability at least $1 - \delta$, we have*

$$\sum_{t=1}^{T}\sum_{h=1}^{H}(\xi_h^t + \zeta_h^t) \le 4\sqrt{TH^3 \log\left(\frac{2}{\delta}\right)}. \tag{55}$$

**Term** $(i)$**:** It turns out that the dominant term and the challenging term to bound is the first term in Lemma 7. We next provide an upper bound on this term.

For step $h$, let $\mathcal{U}_h^T = \bigcup_{t=1}^{T}\mathcal{S}_h^t$ be the union of all cover elements used by $\pi$-KRVI over all episodes. The size of $\mathcal{U}_h^T$ is bounded in the following lemma and is useful in the analysis of Term $(i)$.

**Lemma 9 (Lemma** 2 **in** Janz et al. (2020)**)** *The size of $\mathcal{U}_h^T$ satisfies*

$$|\mathcal{U}_h^T| \le C_6 T^{\frac{d}{d+\alpha}}, \tag{56}$$

*for some constant $C_6$.*

The size of $\mathcal{U}_h^T$ depends on the dimension of the domain and the parameter $\alpha$ used in the splitting rule in Section 3.1.

Now, consider a cover element $\mathcal{Z}' \in \mathcal{U}_h^T$. Using Theorem 1, we have, with probability at least $1 - \delta$, for all $h \in [H], t \in [T], z \in \mathcal{Z}'$, for some $\epsilon_h^t \in (0, 1)$,

$$\left|r_h(z) + [P_h V_{h+1}](z) - \widehat{Q}_h^t(z)\right| \le \beta_h^t(\delta, \epsilon_h^t)b_h^t(z) + \epsilon_h^t, \tag{57}$$

where $\beta_h^t(\delta, \epsilon_h^t)$ is the smallest value satisfying

$$\beta_h^t(\delta, \epsilon_h^t) \ge H + 1 + \frac{H}{\sqrt{2}}\sqrt{\Gamma_{k,\lambda}(N) + \log\mathcal{N}_{k,h}(\epsilon_h^t; R_N, \beta_h^t(\delta, \epsilon_h^t)) + 1 + \log\left(\frac{NH}{\delta}\right)} + \frac{3\sqrt{N}\epsilon_h^t}{\lambda},$$

with $N = N_{h,\mathcal{Z}'}^T$ and $\epsilon_h^t = \frac{H\sqrt{\log(\frac{TH}{\delta})}}{\sqrt{N_{h,\mathcal{Z}'}^T}}$.

We also note that

$$\begin{aligned}
\Gamma_{k,\lambda}(N_{h,\mathcal{Z}'}^T) &= \mathcal{O}\left((N_{h,\mathcal{Z}'}^T)^{\frac{1}{p}}(\log(N_{h,\mathcal{Z}'}^T))^{1-\frac{1}{p}}\rho_{\mathcal{Z}'}^{\frac{\alpha}{p}}\right) \\
&= \mathcal{O}\left((\rho_{\mathcal{Z}'})^{\frac{-\alpha}{p}}(\log(N_{h,\mathcal{Z}'}^T))^{1-\frac{1}{p}}\rho_{\mathcal{Z}'}^{\frac{\alpha}{p}}\right) \\
&= \mathcal{O}\left((\log(N_{h,\mathcal{Z}'}^T))^{1-\frac{1}{p}}\right) \\
&= \mathcal{O}\left(\log(T)\right),
\end{aligned} \tag{58}$$

where the first line is based on Lemma 2, and the second line is by the design of partitioning in $\pi$-KRVI. Recall that each hypercube is partitioned when $\rho_{\mathcal{Z}'}^{-\alpha} < N_{h,\mathcal{Z}'}^t + 1$, ensuring that $N_{h,\mathcal{Z}'}^t$ remains at most $\rho_{\mathcal{Z}'}^{-\alpha}$.

For the covering number, with the choice of $\epsilon_h^t = \frac{H\sqrt{\log(\frac{TH}{\delta})}}{\sqrt{N_{h,\mathcal{Z}'}^t}}$, we have

$$
\begin{aligned}
&\log \mathcal{N}_{k,h}(\epsilon_h^t; R_N, \beta_h^t(\delta, \epsilon_h^t)) \\
&= \mathcal{O}\left( \left( \frac{R_N^2 \rho_{\mathcal{Z}'}^\alpha}{(\epsilon_h^t)^2} \right)^{\frac{1}{p-1}} (1 + \log(\frac{R_N}{\epsilon_h^t})) + \left( \frac{(\beta_h^t(\delta, \epsilon_h^t))^2 \rho_{\mathcal{Z}'}^\alpha}{(\epsilon_h^t)^2} \right)^{\frac{2}{p-1}} (1 + \log(\frac{\beta_h^t(\delta, \epsilon_h^t)}{\epsilon_h^t})) \right) \\
&= \mathcal{O}\left( \left( \frac{R_N^2}{H^2 \log(\frac{HT}{\delta})} \right)^{\frac{1}{p-1}} (1 + \log(\frac{R_N}{\epsilon_h^t})) + \left( \frac{(\beta_h^t(\delta, \epsilon_h^t))^2}{H^2 \log(\frac{HT}{\delta})} \right)^{\frac{2}{p-1}} (1 + \log(\frac{\beta_h^t(\delta, \epsilon_h^t)}{\epsilon_h^t})) \right).
\end{aligned}
$$

We thus see that the choice of $\beta_h^t(\delta, \epsilon_h^t) = \Theta(H\sqrt{\log(\frac{TH}{\delta})})$ satisfies the requirement for confidence interval width on $\mathcal{Z}'$ based on Theorem 1. We now use probability union bound over all $\mathcal{Z}' \in \mathcal{U}_h^T$ to obtain

$$
\beta_T(\delta) = \Theta(H\sqrt{\log(\frac{TH|H\mathcal{U}_h^T|}{\delta})}) = \Theta(H\sqrt{\log(\frac{TH}{\delta})}). \tag{59}
$$

For this value of $\beta_T(\delta)$, we have with probability at least $1 - \delta$, for all $h \in [H], t \in [T], z \in \mathcal{Z}$,

$$
\left| r_h(z) + [P_h V_{h+1}](z) - \widehat{Q}_h^t(z) \right| \le \beta_T(\delta) b_h^t(z) + \epsilon_h^t, \tag{60}
$$

where in the above expression $\epsilon_h^t$ is the parameter of the covering number corresponding to $\mathcal{Z}'$ when $z \in \mathcal{Z}'$.

Therefore, we have, with probability at least $1 - \delta$

$$
\text{Term } (i) \le \sum_{t=1}^T \sum_{h=1}^H -\delta_h^t(z_h^t) \le 2\beta_T(\delta) \left( \sum_{t=1}^T \sum_{h=1}^H b_h^t(z_h^t) \right) + 2\epsilon_h^t, \tag{61}
$$

with

$$
\epsilon_h^t = \frac{H\sqrt{\log(\frac{TH}{\delta})}}{\sqrt{N_{h,\mathcal{Z}(z_h^t)}^t}}. \tag{62}
$$

We bound the total uncertainty in the kernel ridge regression measured by $\sum_{t=1}^T \left( b_h^t(z_h^t) \right)^2$

$$
\begin{aligned}
\sum_{t=1}^T \left( b_h^t(z_h^t) \right)^2 &= \sum_{\mathcal{Z}' \in \mathcal{U}_h^T} \sum_{z_h^t \in \mathcal{Z}'} \left( b_h^t(z_h^t) \right)^2 \\
&\le \sum_{\mathcal{Z}' \in \mathcal{U}_h^T} \frac{2}{\log(1 + 1/\lambda^2)} \Gamma_{k,\lambda}(N_{h,\mathcal{Z}'}^T) \\
&= \mathcal{O}\left( \sum_{\mathcal{Z}' \in \mathcal{U}_h^T} \log(T) \right) \\
&= \mathcal{O}\left( |\mathcal{U}_h^T| \log(T) \right) \\
&= \mathcal{O}\left( T^{\frac{d}{d+\alpha}} \log(T) \right).
\end{aligned}
$$

The first inequality is commonly used in kernelized bandits. For example see (Srinivas et al., 2010a, Lemma 5.4). The third and fifth lines follow from Equation (58) and Lemma 9, respectively. Also,

we have

$$
\begin{aligned}
\sum_{t=1}^{T} (\epsilon_h^t)^2 &= \sum_{t=1}^{T} \frac{H^2 \log(\frac{TH}{\delta})}{N_{h,\mathcal{Z}(z_h^t)}^t} \\
&\leq \sum_{\mathcal{Z}' \in \mathcal{U}_h^T} \sum_{z_h^t \in \mathcal{Z}'} \frac{H^2 \log(\frac{TH}{\delta})}{N_{h,\mathcal{Z}'}^t} \\
&\leq |\mathcal{U}_h^T| H^2 \log(\frac{TH}{\delta})(\log(T) + 1) \\
&\leq \mathcal{O}\left( H^2 T^{\frac{d}{d+\alpha}} \log(\frac{TH}{\delta}) \log(T) \right).
\end{aligned}
\tag{63}
$$

We are now ready to bound the

$$
\begin{aligned}
\text{Term } (i) &\leq 2\beta_T(\delta) \left( \sum_{t=1}^{T} \sum_{h=1}^{H} b_h^t(z_h^t) \right) + 2 \sum_{t=1}^{T} \sum_{h=1}^{H} \epsilon_h^t \\
&\leq 2\beta_T(\delta)\sqrt{T} \sum_{h=1}^{H} \sqrt{\sum_{t=1}^{T} (b_h^t(z_h^t))^2} + 2\sqrt{T} \sum_{h=1}^{H} \sqrt{\sum_{t=1}^{T} (\epsilon_h^t)^2} \\
&= \mathcal{O}\left( H^2 T^{\frac{d+\alpha/2}{d+\alpha}} \sqrt{\log(T) \log(\frac{TH}{\delta})} \right).
\end{aligned}
\tag{64}
$$

The proof is completed.

