# OpenReview forum: "Kernelized Reinforcement Learning with Order Optimal Regret Bounds"
_NeurIPS.cc/2023/Conference — NeurIPS 2023 poster_

### Official Review · Reviewer_o5D2 · 2023-07-01

**Soundness:** 1 poor
**Presentation:** 2 fair
**Contribution:** 2 fair
**Rating:** 6
**Confidence:** 4

**Summary:**

This work studies RL with kernel function approximation, specifically where it is assumed that the transition dynamics and reward function live in some RKHS. While there has been some work on RL with kernel function approximation, existing work provides bounds which are suboptimal. This work seeks to tighten these bounds, and ultimately obtains a bound scaling as $O(\sqrt{\Gamma(T) T})$ where $\Gamma(T)$ is the maximum information gain, matching the lower bound in certain settings of the bandit case. Their proposed algorithm is a variant of optimistic LSVI.

**Strengths:**

1. The setting of kernel RL has not been studied as thoroughly as some areas of RL, and optimal bounds do not exist. This work takes a step in obtaining optimal regret, tightening the best-known existing bounds, and obtaining optimal regret in certain special cases (some bandit instances). However, I believe the stated result is incorrect—see below.

**Weaknesses:**

1. I do not believe the stated result is correct. In the setting of linear bandits/linear MDPs/linear mixture MDPs, the information gain is bounded as $\Gamma_{k,\lambda}(T) \le O(d \log T)$, which would translate to a regret guarantee scaling as $O(\sqrt{d T})$. However, there are well-known lower bounds for these settings which scale as $\Omega(d \sqrt{T})$ (see e.g. [1] and [2]). Thus, the bound stated in this paper is better than the lower bound by a factor of $\sqrt{d}$, which is impossible.

The following are less significant issues, but are also important to address:

2. There is a vast body of literature on RL beyond tabular and linear function approximation which is not referenced or discussed. See references [3-6] below for a start, and the works cited therein. This literature should be cited and discussed.
3. To make the results concrete, it would be helpful to instantiate Theorem 2 in the setting of tabular and linear MDPs.
4. The result is only provably optimal in the setting of bandits with Matern kernels. However, this paper considers deterministic rewards, so it doesn’t actually capture the bandit setting and as such does not handle the setting for the stated lower bound. It’s difficult, then, to make the claim that this result is optimal in any setting. I would suggest modifying the setting to allow for noisy rewards, or showing that there is a reduction from the current setting to the bandit setting (by encoding reward randomness in the transitions).
5. In addition, it would greatly strengthen the paper to show a lower bound for kernelized RL.
6. It was not clear to me what the $\eta$ parameter corresponded to or how it was defined. This should be clarified.
7. A reference or proof should be given for Lemma 1.

[1] Zhou, Dongruo, Quanquan Gu, and Csaba Szepesvari. "Nearly minimax optimal reinforcement learning for linear mixture markov decision processes." Conference on Learning Theory. PMLR, 2021.

[2] Lattimore, Tor, and Csaba Szepesvári. Bandit algorithms. Cambridge University Press, 2020.

[3] Du, Simon, et al. "Bilinear classes: A structural framework for provable generalization in rl." International Conference on Machine Learning. PMLR, 2021.

[4] Jin, Chi, Qinghua Liu, and Sobhan Miryoosefi. "Bellman eluder dimension: New rich classes of rl problems, and sample-efficient algorithms." Advances in neural information processing systems 34 (2021): 13406-13418.

[5] Foster, Dylan J., et al. "The statistical complexity of interactive decision making." arXiv preprint arXiv:2112.13487 (2021).

[6] Zhong, Han, et al. "A posterior sampling framework for interactive decision making." arXiv preprint arXiv:2211.01962 (2022).

**Questions:**

Please provide further clarification on the instantiation of the results to linear bandits/MDPs (see comment above).

**Limitations:**

Yes.

---

> ### Author Rebuttal · Authors · 2023-08-07
>
> Thank you for reviewing our paper and providing helpful comments. We have addressed your questions below and hope that it will positively affect your evaluation of the paper. We would also like to highlight that we firmly disagree that our results contradict the lower bounds for linear case, as clarified below. We welcome further discussions if required.
>
> 1. Our results do not contradict the lower bounds for linear case, as mentioned in the review. Several NeurIPS, AISTATS and ICML published works on kernel bandits, including [20,21,40], report $\mathcal{O}(\sqrt{\Gamma(T) T})$ regret bounds with $\Gamma(T)$ as the maximum information gain. The reasoning in the review would challenge all such results, which is not correct. The $\mathcal{O}(\sqrt{\Gamma(T) T})$ regret bounds match the lower bounds given in [19] for kernel-based bandits with Matérn kernel. The reasoning in the review seem to suggest that $\mathcal{O}(\Gamma(T)\sqrt{ T})$ should be the best achievable regret bounds in kernel bandits/RL. This, however, is undesirable as it may be trivial (superliner). Also see [18] for a technical discussion on the challenges in desining algorithms obtaining $\mathcal{O}(\sqrt{\Gamma(T) T})$ regret bounds.  This comment from the reviewer further emphasizes the significance of our results and our contribution to the literature.
>
> The alleged contradiction between the $\mathcal{O}(\sqrt{\Gamma(T) T})$ upper bounds and the $\Omega(d\sqrt{T})$ lower bound in the linear case arises from an oversight. The upper bound $\mathcal{O}(\sqrt{\Gamma(T) T})$ is more explicitely  $\mathcal{O}(\sqrt{d\Gamma(T) T})$, with an $\sqrt{d}$ factor hidden in the $\mathcal{O}$ notation. When applied to linear case, we recover $\mathcal{O}(d\sqrt{ T})$ regret as expected. In kernel settings, $d$ and $\Gamma(T)$ represent different notions of dimension: $d$ is dimension of the input space, while $\Gamma(T)$ , roughly speaking, represents the effective dimension of the kernel. Although they coincide in the linear case, in general kernel-based settings, $\Gamma(T)$, which grows with $T$, is the dominating term. Hence, it is customary to omit the $d$ terms in the expression of regret bounds in kernel-based bandit and RL literature, e.g., [10, 16, 18, 20, 21, 40]. See a similar discussion in the paragraph next to last in [18]. We will add the $d$ term in our final expression of regret bound and explain this point in the paper, also commenting on the linear case.
>
> We hope this clarification addresses the concern. We welcome further discussion and suggest referring to [18, 20] for a detailed technical exploration of this intricate issue.
>
> 2. We acknowledge the extensive RL literature. We have cited crucial related works, including [10] and references therein, as well as relevant literature on bandits. We value the reviewer's recommendations and will incorporate the suggested references, discussing their relevance and contributions.
>
> 3. In the tabular setting, the regret bound scales with the cardinality of state-action space. Thus, when the cardinality of state action space is very large, we obtain tighter regret bounds. Our regret bounds remain the same for a finite state-action space. For the linear model, please see response to question 1.
>
> 4. Thanks for mentioning this point. Even with deterministic rewards, the observations of target function are noisy due to the random transition to the next step. While we agree that noisy rewards makes the problem more general and aid in comparisons with noisy bandits, they will not effectively change the results in general. Hence, deterministic rewards are often assumed for simplicity, as seen in [10], without loss of generality. Reference [10] also compares the results with the noisy bandit setting. We value your feedback and will incorporate this discussion in the final version.
>
> 5. Having a general lower bound that incorporates the episode length $H$ seems a significantly challenging problem, which is beyond the scope of this paper.
>
> 6. The introduction of the parameter $\eta$ is a technical formality for the correctness of the results, as also given in [10]. In some works, $\eta$ is simply assumed to be zero by assuming the Mercer eigenfeatures are uniformly bounded. This, however, may not hold in general as Mercer eigenfeatures depend on the shape of the domain. We introduce $\eta$ such that $\sigma_m^{\eta}\phi_m$ are uniformly bounded. This uniform boundedness is used in the bounds on maximum information gain and function calss covering number. The uniform boundedness of $\sigma_m^{\eta}\phi_m$ holds for regular domains such as hyperspheres and hypercubes with typical kernels such as Matérn [10].
>
> 7. This result is also used in [10]. Following your comment, we will add a proof (e.g., proof of Lemma 3 in Yeh et al., AISTATS'23).
>
> S. Y. Yeh, F. Chang, C. Yueh, P. Wu, A. Bernacchia, S. Vakili, "Sample Complexity of Kernel-Based Q-Learning", AISTATS 2023.

---

> > ### Comment · Reviewer_o5D2 · 2023-08-15
> > **Reply to rebuttal**
> >
> > I would like to thank the authors for their detailed response. I had a few follow-up questions:
> >
> > 1. Could the authors point me to the specific line in the proof where the factor of $d$ is hidden?
> >
> > 4. I agree that considering stochastic rewards should not change the results, and that it is often possible to embed a bandit in a 1-step RL problem with deterministic reward by encoding the bandit reward noise into the stochastic transition. As such, I agree with the authors that it's not unreasonable to compare with the bandit lower bound even when considering deterministic rewards. However, I do think this should be mentioned explicitly in the discussion around equation (22) (or the embedding of bandits into 1-step RL problems with deterministic rewards made explicit).

---

> > > ### Author Response · Authors · 2023-08-16
> > > **Response to follow up questions**
> > >
> > > We greatly value the reviewer's engagement and helpful comments.
> > >
> > > In response to follow up questions:
> > >
> > > 1. Regarding the input dimension $d$: The constatns in front of both the information gain and the logarithm of the covering number scale with $d$. Consequently, the confidence interval width multiplier $\beta_h^t(\delta, \epsilon)$, as given in Equation (19)), also increases with $d$. This leads to the constants in regret bound increasing with $d$. The growth of constants in both the information gain and the logarithm of the covering number is attributed to the constants in the kernel's spectrum increasing with $d$. Specifically, within the proof of Lemmas 2 and 3, these constants scale with $d$ (in Equations (39), (47) and the subsequent one, and (52) and the one follwoing). In proof of Theorem 2, this scaling impacts the constant in Equation (59) and the implied constant, in $\Theta$ notation, in Equation (60). That carries to the constant in Equation (65), which appears in the final regret bounds.
> > >
> > > We wish to highlight that hiding constants dependent on $d$ is not a peculiarity of our presentation. Such a practice is commonplace in all kernel bandits and RL papers we are familiar with, as evident by well-cited papers such as [10, 14, 16, 20, 21, 22, 31, 40, 44, 45]. In this line of work, the primary focus has been on the regret growth with $T$, stiving for sublinear and, ideally, order-optimal regret bounds in $T$, that has overshadowed attention to $d$. Prompted by the reviewer's comment, we will include a more detailed explanation and clearer discussion on this point in the final version.
> > >
> > > 2. We completely agree with this comment. We will elaborate on this point in the final version of the paper. Specifically, we will clarify that adding observation noise to rewards does not affect the order of regret bounds presented. Therefore, we can compare our results to the noisy bandits, as in [10], without any loss of generality.
> > >
> > > We appreciate the reviewer's constructive feedback. We are confident that incorporating these discussions will enrich our exposition, further improving the quality of our paper.

---

> > > > ### Comment · Reviewer_o5D2 · 2023-08-16
> > > > **Response**
> > > >
> > > > I spent some time going through the proof more carefully and agree with the authors comments. As such, I will increase my score accordingly.
> > > >
> > > > I would strongly encourage the authors to make this more transparent, however. In particular, there are other terms hidden in absolute constants and big-O notation (such as $C_p$), which is only mentioned deep in the proofs. The big-O notation should be formally defined so it is obvious from reading the text in the main body what is being hidden.

---

> > > > > ### Author Response · Authors · 2023-08-18
> > > > >
> > > > > Thank you. We greatly appreciate your participation in the discussion and the valuable feedback. We will include the aforementioned discussions on noisy rewards, as well as dependence of constants on $d$ and $p$, and the connection to linear case in the final version. These additions will strengthen the overall quality of the paper.

---

### Official Review · Reviewer_2xmH · 2023-07-09

**Soundness:** 3 good
**Presentation:** 3 good
**Contribution:** 3 good
**Rating:** 6
**Confidence:** 3

**Summary:**

This paper proposes a reinforcement learning algorithm called $\pi$-KRVI that achieves order-optimal regret. The algorithm performs local kernelized optimistic least squares value iteration update - specifically, it partitions the state-action space so that each cell contains a small number of observations, and the Q value within a cell is updated to a Gaussian Process based Upper Confidence bound using observations in that cell only. This is motivated by KOVI [10] and $\pi$-GP-UCB [14]. A sublinear upper bound on the regret is given - this seems to be the first sublinear bound in a general setting, and it is claimed to be order optimal in the number of episode $T$.

**Strengths:**

The sublinear regret seems to be the first one established in a general setting, and this seems to be order-optimal. The paper is generally clear, but a few things could be improved as mentioned in Weaknesses. The idea seem to be sound, but I didn't read the proofs.

**Weaknesses:**

* The paper only informally refers to [19] when stating the bound is order optimal. This is not immediately clear due to differences in notations, and some ambiguity in the paper's discussion. A more precise and detailed discussion would be helpful.
* It is not clear what ${\cal S}$ and ${\cal A}$ are, but it seems both are cubes? If yes, is this a necessary assumption?
* The reward functions and transition distributions are assumed to have a norm $\le 1$. How will the regret change if the upper bound is larger than 1?
* Lemma 1: $V$ is not used anywhere. Is it supposed to be $V_{h+1}$? If yes, an arbitrary $V_{h+}$ can potentially have a large norm, and Eq. (11) may not hold? A proof of the lemma seems to be missing.
* Line 233: why is the target value constructed using a new random state $s_{h+1}'$, rather than the random state $s^{t}_{h+1}$ from the current episode?
* Line 248: Does the running time refer to the total running time of Algorithm 1? If yes, can you explain why the upper bound is independent of the size of the partitions?

Minor comments
* Kernel ridge regression is generally described as a method that doesn't provide any uncertainty estimate, while the paper describes it as a method that provides uncertainty estimates. In particular, [38] is cited as the reference, but it doesn't seem to provide such an account of kernel ridge regression. What's described as kernel ridge regression is Gaussian process regression.
* A brief explanation on the motivation behind maximum information gain would be helpful.
* Putting the pseudocode of the algorithm in the paper would be helpful. Alternatively, provide a more complete description of the algorithm.
* Algorithm 1, line 10: $x_{h}^{t}$ should be $s_{h}^{t}$?

**Questions:**

I would appreciate a discussion on the weaknesses before minor comments.

**Limitations:**

A discussion on the limitation of Assumption 1 would be beneficial.

---

> ### Author Rebuttal · Authors · 2023-08-07
>
> Thank you for reviewing our paper. We appreciate your feedback and we are glad that you find the paper generally clear. We respond to all your comments in the weaknesses section (in order) and hope that it positively affects your evaluation of the paper.
>
> - In the kernel based bandit setting, which corresponds to the special case of $H=1$ and $|S|=1$, [19] established a lower bound of $\Omega(T^{\frac{\nu+d}{2\nu+d}})$. This lower bound implies that the regret bound in Theorem 2 cannot be improved in terms of $T$, in general. We however agree with the reviewer that a lower bound under the RL setting, taking into account the general case and incorporating $H$ would be useful. That however seems itself a challenging problem and beyond the scope of this paper. We will add a more detailed and precise discussion following your comment.
>
> - In our construction of the domain partitioning, we assume that $\mathcal{S}\times \mathcal{A}$ is a hypercube, implying $\mathcal{S}$ and $\mathcal{A}$ are hypercubes. This is for simplicity and correctness of a formal derivation. This assumption is not necessary for the overall approach. The domain partitioning technique can be applied to other compact subsets of $R^d$ by considering the smallest cube containing the domain. This allows us to apply the partitioning technique and obtain valid results for RL problems with non-cube-shaped state and action spaces.
>
> - Scaling the norms of the reward functions and transition distributions with a constant 'C' would scale the "H+1" term in the confidence interval (see Theorem 1) with C. This represents the RKHS norm of the target function in kernel regression. However, this additive term does not affect the order of the regret bounds, as it is overshadowed by the dominant terms (information gain and log covering number), which grow with $T$. This is similar to the case of kernel bandits, where the regret order in $T$ remains unchanged as long as a bound on the RKHS norm of the target function is known. Only if the RKHS norm of the function scales with $T$ would further investigation be needed to understand the impact on the regret bounds.
>
> - Thanks for mentioning this. There is a typo and $V_{h+1}$ should be $V$. We express the lemma for a general $V:\mathcal{S}\rightarrow [0,H]$. This result is also used in [10]. We will add a proof (e.g., see Lemma 3 in Yeh et al., AISTATS'23 for a proof).
>
> S. Y. Yeh,  F. Chang, C. Yueh, P. Wu, A. Bernacchia, S. Vakili, "Sample Complexity of Kernel-Based Q-Learning", AISTATS 2023.
>
> - We believe the notation is consistent and correct. The episode index $t$ is specified in the notation $Z_h^t$ defined in Line 230.
>
> - Yes, it is the total running time of Algorithm 1. At each episode $t$ and each step $h$, the computation of the kernel ridge regression statistics in each hypercube has a cost of $O(N_c^3+|A_c|N_c^2)$ where $|A_c|$ is the number of actions in the hypercube and $N_c$ is the number of previous observations in the hypercube. Summing up over all hypercubes, we bound the computational complexity with $O(t^3+|\mathcal{A}|t^2)$, where $|\mathcal{A}|$ is the total number of actions. This bound is obtained using the simple arithmetic that the cube of the sum of natural numbers is larger than the sum of their cubes. Summing up over steps and episodes, we arrive at the overall runtime complexity of $O(HT^4+H\mathcal{A}T^3)$. This calculation is similar to [14], and analogous to [14], we expect an improved runtime for $\pi$-KRVI in practice due to inequalities used in this calculation. We will add further details and clarification on this calculation following your comment.
>
> *Minor comments*
>
> - Kernel ridge regression and Gaussian process (GP) regression lead to essentially the same calculation for the prediction and uncertainty estimate based on data, but from different Bayesian and frequentist perspectives, respectively. The equivalence between these two methods and an interpretation of the uncertainty estimate within kernel ridge regression framework are discussed in detail in Chapter 3 of "Gaussian Processes and Kernel Methods: A Review on Connections and Equivalences", M Kanagawa, P Hennig, D Sejdinovic, and B K Sriperumbudur. We find the terminology of kernel ridge regression better suited for our case, similar to [10], as the use of GP terminology (when a *surrogate* GP is employed for modeling the target function) may incorrectly imply that the target function is a sample from a GP. We will further clarify this point and cite Kanagawa et al. as a reference.
>
> - Roughly speaking, information gain represents the effective dimension of the kernel. That is to say, while typical kernels have an infinite-dimensional feature space, only a limited number of features have a significant effect on the regression with a finite dataset. This represents the effective dimension that is the same as information gain up to a log factor. This quantity appears in the confidence intervals for kernel-based models as shown in the results on vector valued self-normalized martingale inequalities and their extension to Hilbert spaces [17,34,42]. We will add this explanation to the final version of the paper.
>
> - The pseudocode of $\pi$-KRVI is provided in Appendix A in the submitted version of the paper. Due to space limitations, we have placed it in the appendix. However, in the final version of the paper, with the availability of an extra page, we will include the pseudocode in the main body of the paper.
>
> - Thanks for pointing out the typo. We will correct it.

---

### Official Review · Reviewer_fuZF · 2023-07-21

**Soundness:** 4 excellent
**Presentation:** 4 excellent
**Contribution:** 4 excellent
**Rating:** 7
**Confidence:** 4

**Summary:**

This paper presents an optimism-based online learning algorithm for RL with large state-action spaces (including continuous spaces). It proposes a (Gaussian) kernel-based function approximation + optimism (building on UCBVI) algorithm. It assumes that the reward and transition density functions are representable in  1-bal of an RKHS (with a Gaussian kernel), which is quite reasonable.  It also introduces a domain-partitioning technique  to make the kernel ridge regression part scalable. The regret bound obtained for the algorithm is shown to be an improvement over SOTA [10]. Specifically, regret scales H^2 and sublinear in T.

**Strengths:**

Online RL algorithms for continuous state and action spaces is a really challenging problem, and until recently was unresolved. This is the best such result I have seen. It makes a very smart (and now seemingly natural) use of kernel-based function approximation.

**Weaknesses:**

The authors have not presented any numerical results. So it leaves one wondering whether it is all nice theory, and there is some hope of the making the algorithms practical.

Note: Title has a typo for Kernelized".

**Questions:**

1. Could you present some numerical studies so we can understand the strength of your algorithm? It could even be large tabular but difficult problems such as DeepSea and Montezuma's revenge.

**Limitations:**

Kernel-based method may limit scalability.

---

> ### Author Rebuttal · Authors · 2023-08-07
>
> Thank you for reviewing our paper. We appreciate your feedback and are glad that you mention the significance and novelty our results.
>
> While we agree that numerical experiments are important for evaluating the practicality of RL algorithms, our main contribution in this paper is theoretical in nature, similar to existing work, e.g., [9-10].
>
> There seems to be a natural progression of model complexity in RL, from tabular to linear, to kernel-based, and to deep learning based. While tabular and linear problems are adequately studied, the state of the art delas with kernel-based RL. Our work makes a significant contribution by providing an order-optimal regret bound in the number of episodes for a broad class of common kernels. This contribution is particularly important considering that existing results fail to demonstrate even sublinear regret bounds for this class of kernels. We appreciate your recognition of the significance of our results.

---

> > ### Comment · Reviewer_fuZF · 2023-08-21
> >
> > I am not convinced that a numerical investigation here won't be useful. With kernel methods, scalability is a key concern. I feel my original score was generous, and I will keep it.

---

### Official Review · Reviewer_qSio · 2023-07-29

**Soundness:** 3 good
**Presentation:** 3 good
**Contribution:** 2 fair
**Rating:** 5
**Confidence:** 3

**Summary:**

This paper theoretically studies the performance of a reinforcement learning algorithm under the assumption that the Q function is a member of RKHS with a known kernel. The authors provide cumulative regret bound on the iterative-least value iteration algorithm and specialise their results to the kernel with polynomial decay of eigenvalues. They improve the regret bounds of prior works by refined analysis of confidence sets in this non-parametric setting.

**Strengths:**

The paper is clearly written in the context of its rather technical nature.

The work combines the newest understanding of adaptive confidence sets and their analysis for the case of Matern kernels with Linear MDPs, by providing a kernelized variant of thereof. I am not familiar with the other literature utilising the novel variants of the analysis in the context of RL.

Seems like there is a sub-community interested in this issue given the COLT open problem, however, as somebody not in this community, I have to say it seems certainly a matter of taste rather than importance.


**Weaknesses:**

The prior work, [10] indeed does not provide optimal bounds, but arguably the algorithm seems to be more practical than a rather time-varying discretization of the domain in order to facilitate the construction of the order optimal confidence sets.

To be slightly harsh I wonder if the proper academic solution would not have been informing the authors of [10] of this new technique rather than writing a wholly new paper which utilises the trick with the domain splitting which comes from other prior works e.g. [14]. At the expense of creating more elaborate confidence sets one can indeed improve the bounds, but whether this improves the performance remains unanswered in this paper, as no comparison is provided. The proposed contribution is solely of theoretical nature.

It is common in online learning to use the doubling trick and mention it in passing in case improved results are desired. I wonder if this is not a similar discretization trick, and whether this deserves an independent publication at NeurISP.


**Questions:**

- Mercer decomposition diagonalize the infinite dimensional operator either under a specific distribution or on a bounded domain (essentially uniform distribution), could you be more specific what you mean by the Mercer decomposition in Section 2.2
- Why is there H^2 in the regret bounds? What is the nature of this? One "H" would be more intutivie.
- How are quantities Infogain and covering numbers related?
- Does one also need info-gain to derive the bound? Clearly there are bound with only the info-gain, can there be only covering number bounds? I wonder if works by van der Geer on high dimensional statistics are related to this.
- Covering numbers for Sobolev paper are well-studied objects in functional analysis, especially due to seminal paper of field’s medalist Smale. I suggest looking into ref in publications of this author.


**Limitations:**

None identified.

---

> ### Author Rebuttal · Authors · 2023-08-07
>
> Thank you for reviewing our paper. We value your feedback and are pleased that you found the paper clearly written. We address your comments below and hope that it will positively impact your evaluation of the paper. We are open to further discussion if required.
>
> In terms of contribution, we would like to mention that we make substantial contribution to the related work. For example, we provide flexibility in setting the parameters of the confidence interval (in Theorem 1), that ultimately contributes to the improved regret bounds. We also derive bounds on the maximum information gain (Lemma 2) and the function class covering number (Lemma 3), taking into consideration the size of the state-action domain. Lemma 2 provides a stronger bound than the one used in [14], which also contributes to the improved regret bounds. Thus, in applying the partitioning technique of [14] to the kernel-based RL problem, we make significant improvements to the technique and provide a finer analysis that also significantly improves the results of [14], as stated in lines 91-93 in the introduction. To further clarify this point, we will report the regret bounds for kernel bandits in [14, Theorem 3], which is in order of $\tilde{O}(T^{(d(2d+3)+2\nu)(d(2d+4)+4\nu)})$, and has a polynomial gap from the $\mathcal{O}(T^{(\nu+d)/(2\nu+d)})$ proven in our work for the more involved RL setting. Given these significant improvements, we believe our results constitute a significant contribution to the literature regarding achievable regret bounds in kernel-based RL. Nonetheless, our work reports the first order optimal regret bounds in $T$ for a broad class of common kernels, which is a novel result that may be of wide interest in the community and become a main reference for the achievable regret bounds.
>
> Responses to all questions (in order):
>
> -  A formal presentation of Mercer theorem with the details is provided in Appendix B. For example, we can consider the Lebesgue measure as stated in Appendix B. We note that our analysis only relies on the expressions of kernel and RKHS elements in terms of eigenvalues and eigenfeatures given in equations (7) and (8). The choice of measure in Mercer theorem only affects the eigenvalues and eigenfeatures, and not the expressions in equations (7) and (8) or the rest of our proof. For regular domains such as hyperspheres and hypercubes, with typical kernels, Mercer decompositions are well studied and can be used within our results. Following your comment, we will make this further clear in the paper.
>
> - One $H$ scaling seems to be due to the RKHS norm of state-action value functions, the target functions in kernel ridge regression. Lemma 1 shows that this norm scales with $H$. Our results on regret bound align with those in the SOTA [10]. With our results, the optimality of scaling with $H$ cannot be determined. We will make this point further clear in the paper.
>
> - Information gain and covering number both represent the complexity of the function class that we are considering. Roughly speaking, information gain represents the effective dimension of the kernel. That is to say, while typical kernels have an infinite-dimensional feature space, only a limited number of features have a significant effect on the regression with a finite dataset. This represents the effective dimension that is the same as information gain up to a log factor. The covering number, on the other hand, is determined by the RKHS topology, gauging the count of RKHS elements required to approximate the entire set within a given sup norm error margin.
>
> While information gain and covering number represent distinct complexities of the problem, they are both characterized using the kernel spectrum. Thus, it is expected that their value is related. Please see Lemmas 2 and 3 for bounds on their value. In our domain partitioning algorithm, we set $\epsilon$ proportional to $1/\sqrt{N}$, where $N$ is the number of samples in the corresponding subdomain, while [10] sets $\epsilon$ proportional to 1/T. In both cases, covering number turns to be the dominating term in the confidence intervals width factor.
>
> - We here provide a brief explanation on where these quantities appear in the analysis. Please let us know if further details are needed.
>
> One important component in the analysis is the confidence intervals for the kernel-based models. The information gain appears due to adaptivity of samples and the covering number appears to to variation of the target function. In an offline setting, $1-\delta$ confidence intervals of the form $|f(x)-\mu_t(x)| \le \beta(\delta)\sigma_t(x)$ are established where $\beta(\delta) = R+\sqrt{\log(\frac{1}{\delta})}$ up to multiplicative constants, which are not related to information gain and covering number. In the online setting, with adaptively collected samples (bandit or RL, for example), information gain appears in the confidence interval $\beta(\delta) = R+\sqrt{\Gamma + \log(\frac{1}{\delta})}$. Please see the work on vector valued self-normalized martingale inequalities and their extension to Hilbert spaces [17,34,42]. In addition, in the RL setting, the target function itself is not predetermined and varies due to Markovian dynamics. The confidence interval thus needs to hold uniformly for all functions within a certain class. This is where the log covering number appears in the confidence interval, $\beta(\delta) = R+\sqrt{\log N + \Gamma + \log(\frac{1}{\delta})}$. It is thus expected for both terms to appear in the regret bound of standard LSVI [10]. In our approach with the domain partitioning technique, we ensure both terms stay logarithmic in $T$ at most, and instead we bound the number of partitions, which leads to the improved regret bounds.
>
> - Thank you. We will look into this for future research on the topic.

---

> > ### Comment · Reviewer_qSio · 2023-08-14
> > **response**
> >
> > Thank you for your responses. I liked the clarifications.
> >
> > - Are you the first work to discuss the covering approach for order-optimal confidence sets?
> >
> > - [10] sets covering using 1/N instead of 1/\sqrt{N}. What would their bounds be if they used your discretization with 1/\sqrt{N}?
> >
> > However, there is still a fundamental disagreement in terms of contribution. I feel like you use a trick from paper A [and arguably polish the analysis a bit] to apply in a setting of paper B, where they use it as a tool, to get a *theoretically* better algorithm at the expanse of being more difficult to implement. It is not the same method as in [10] since the confidence construction is more elaborate.  If A and B were completely different fields, I would be fine with this setup, but this essentially the same field. I do not find this surprising that one can use these techniques in the RL context compared to Bandits. These improved confidence sets can be applied and improve the bounds anywhere they need.
> >
> > I am not saying your work does not deserve audience, but, the following description still were accurately characterizes my general opinion of this paper:  "Technically solid paper where reasons to reject, e.g., limited evaluation, outweigh reasons to accept, e.g., good evaluation."
> >
> > I do not share the viewpoint of the other reviewer who claims some lower bounds are contradicted. I think the results are believable but I did not check the maths. They are especially believable since somebody already did near identical analysis before in a different context.

---

> > > ### Author Response · Authors · 2023-08-14
> > >
> > > We greatly appreciate your participation in the discussion.
> > >
> > > Regarding the questions:
> > >
> > > - The use of a covering set to establish confidence intervals applicable to all members of a function class is a requirement imposed by the MDP framework. The target function $r_h+[P_h V_{h+1}^{t}]$ is not fixed and determined; it depends on $V_{h+1}^{t}$due to the temporal dependence in the MDP setting. The utilization of a covering set is also employed in the most related work [10]. However, in establishing our confidence intervals, we allow flexibility in choosing the parameter of the covering set, which ultimately contributes to improved regret bounds with an appropriate parameter selection.
> > >
> > > - The proof in [10] does not provide flexibility in choosing the parameter of the covering set. So we cannot directly observe the answer to your question by setting $\epsilon^*$ (in their notation) in their Theorem 4.2 proportional to $1/\sqrt{t}$, where $t$ is the episode number. However, we can see that the regret bound would at least scale with $O(\Gamma(T)\sqrt{T})$. The reason for this is that the confidence interval width would at least scale with $\sqrt{\Gamma(T)}$, and following the rest of the proof of [10] (which is independent of $\epsilon^*$) leads to a regret bound scaling at least with $O(\Gamma(T)\sqrt{T})$.
> > >
> > > We agree with the reviewer that our improved and order optimal regret bounds come at the price of a more sophisticated algorithm. However, we still believe that our results are significant and of interest to the wider research community for several reasons.
> > >
> > > There is a broad interest in RL and its analysis. We provide the first order-optimal regret bounds in the kernel-based RL setting for a broad class of common kernels. The SOTA fails to show even sublinear regret bounds.
> > >
> > > Our results are not achieved by simply applying a technique from [14] to [10]. Although, we agree that our work is highly inspired by [10] and [14], and we have acknowledged this throughout the paper. Specifically, even in the much simpler problem of kernel bandits, [14] obtained sub-optimal regret bounds of $\tilde{O}(T^{(d(2d+3)+2\nu)/(d(2d+4)+4\nu)})$, while we obtain optimal regret bounds of $\tilde{O}(T^{(\nu+d)/(2\nu+d)})$. This is achieved based on the merit of our Lemma 2, which provides a tighter bound on information gain than the one used in [14], and subsequent improvements to the algorithm and its analysis.
> > >
> > > Also, we would like to address the comment in the review that states "*These improved confidence sets can be applied and improve the bounds anywhere they need*." This statement is not entirely accurate.  We would like to emphasize that the algorithm and domain partitioning are closely intertwined. Domain partitioning alone cannot be used to achieve tighter confidence intervals in general. Instead, a careful and elaborate algorithm that leverages domain partitioning is required to improve the regret bounds. This observation is also highlighted in the recent work of [Lattimore, COLT'23] on the kernel-based confidence intervals, where it is stated that "... any analysis of linear contextual bandits aimed at proving a similar result [order optimal regret bounds] cannot completely decouple the concentration analysis and the algorithm. The same is true for kernelised bandits where the dimension-dependence arising from loose confidence bounds is especially pernicious and can be the difference between sublinear and linear regret".
> > >
> > > Tor Lattimore, "A Lower Bound for Linear and Kernel Regression with Adaptive Covariates'', COLT 2023.
> > >
> > > We hope that these further clarifications improve the reviewer's evaluation of the paper.

---

### Decision · Program_Chairs · 2023-09-21

**Decision:**

Accept (poster)

**Comment:**

This paper provides novel regret bounds for RL under the assumption that the action-value functions are members of an RKHS with a known kernel. This is a well-motivated problem formulation and the authors' results and techniques are sound. The reviewers are, for the most part, supportive of accepting this paper. I support this recommendation, but strongly suggest that the authors incorporate some important writing-oriented suggestions made by the reviewers, notably:
- clarify technical novelty of the approach over the related paper [10].
- make explicit all dependences on the data dimension d and clarify how these results recover the known results for linear function approximation.